

# Generating reliable estimates of tropical cyclone induced coastal hazards along the Bay of Bengal for current and future climates using synthetic tracks

Tim Leijnse[1], Alessio Giardino[1], Kees Nederhoff[2], Sofia Caires[1]

[1] Deltares, Delft, 2600MH, The Netherlands
[2] Deltares USA, 8601 Georgia Ave., Silver Spring, MD 20910, USA

*Correspondence to*: Tim Leijnse (Tim.Leijnse@deltares.nl)

**Abstract.** Deriving reliable estimates of design water levels and wave conditions resulting from tropical cyclones is a challenging problem of high relevance for, among others, coastal and offshore engineering projects and risk assessment studies. Tropical cyclone geometry and wind speeds have been recorded for the past few decades only, therefore resulting in poorly reliable estimates of the extremes, especially at regions characterized by a low number of past tropical cyclone events. In this paper, this challenge is overcome by using synthetic tropical cyclone tracks and wind fields generated by the open source tool TCWiSE (Tropical Cyclone Wind Statistical Estimation), to create thousands of realizations representative for 1,000 years of tropical cyclone activity for the Bay of Bengal. Each of these realizations is used to force coupled storm surge and wave simulations by means of the processed-based Delft3D Flexible Mesh Suite. It is shown that the use of synthetic tracks provides reliable estimates of the statistics of the first-order hazard (i.e. wind speed) compared to the statistics derived for historical tropical cyclones. Based on estimated wind fields, second-order hazards (i.e. storm surge and waves) are computed. The estimates of the extreme values derived for wind speed, wave height and storm surge are shown to converge within the 1,000 years of simulated cyclone tracks. Comparing second-order hazard estimates based on historical and synthetic tracks show that, for this case study, the use of historical tracks (a deterministic approach) leads to an underestimation of the mean computed storm surge up to -30%. Differences between the use of synthetic versus historical tracks are characterized by a large spatial variability along the Bay of Bengal, where regions with a lower probability of occurrence of tropical cyclones show the largest difference in predicted storm surge and wave heights. In addition, the use of historical tracks leads to much larger uncertainty bands in the estimation of both storm surges and wave heights, with confidence intervals being +80% larger compared to those estimated by using synthetic tracks (probabilistic approach). Based on the same tropical cyclone realizations, the effect that changes in tropical cyclone frequency and intensity, possibly resulting from climate change, may have on modelled storm surge and wave heights were computed. An increase in tropical cyclone frequency of +25.6% and wind intensity of +1.6%, based on literature values, could result in an increase of storm surge and wave heights of +11% and +9% respectively. This suggest that climate change could increase tropical cyclone induced coastal hazards more than just the actual increase in maximum wind speeds.



**Keywords**: Tropical cyclones, extreme events, coastal hazards, climate change, Bay of Bengal, synthetic tracks, TCWiSE,
Delft3D FM.

## 1 Introduction

Tropical cyclones (TCs) are among the most destructive natural hazards worldwide. Over the last two centuries, it is estimated that 1.9 million people have lost their lives as a result of TCs worldwide (Shultz et al., 2005; Nicholls et al., 1995). While only about 7% of the global TC form in the Indian oceans, associated damages and casualties surrounding this ocean basin are much

larger than in any other region. Between 1960-2004 it is estimated that more than half a million inhabitants of Bangladesh died because of TC (Shultz et al., 2005). The recent cyclone Amphan (2020) showed that strong TCs still occur in the Bay of Bengal (BoB), where the direct impact has however been mitigated through early warning systems, cyclone shelters and embankments.

A challenge in several coastal engineering applications consists in the determination of reliable estimates of design water levels

and wave conditions resulting from these TC, both for present and future climate scenarios. The estimation of design values resulting from TCs is often based on a limited number of recorded Historical Tropical Cyclones (HTC) at a given region (for the BoB see e.g. Chiu and Small, 2016; Dube et al., 2009). This results in a large statistical uncertainty in estimating the first-order hazards resulting from TCs (e.g. wind speeds) due to the limited number of observations at a certain location. One approach to overcome this is by using Synthetic Tropical Cyclones (STC) based on the statistics of the properties of observed

HTC. The Tropical Cyclone Wind Statistical Estimation tool (TCWiSE; Nederhoff et al., 2021) can, for example, be used to generate numerous synthetic tracks. This allows for the creation of a much longer dataset to perform extreme value analysis on than otherwise available through HTC tracks only, and which can be used to calculate more reliable estimates of first-order hazards. Subsequently, these STC can then be modelled using hydrodynamic and wave models to generate better estimates of second-order hazards like storm surge and wave heights. Other datasets and methods to generate STC are those of Vickery et

al., 2000; Hardy et al., 2003; James and Mason, 2005; Emanuel et al., 2006; Haigh et al., 2014; Lee et al., 2018 and Bloemendaal et al., 2020. However, so far literature has mainly focused on deriving STC tracks and corresponding wind and pressure fields, rather than exploring the effect of using these to derive local design values for storm surge and wave heights compared to considering HTC only. Some work has been done in this direction but with focus on different regions than the BoB (e.g. Australia (Haigh et al., 2014) or USA (Lin et al., 2012, Marsooli et al. 2019)) and/or without taking waves into

account, which is found to be an important factor leading to flooding in the northern BoB (Krien et al., 2017) and arguably worldwide.

The effects that climate change and global warming have on TCs is subject to scientific debate. As discussed in Knutson et al. (2010), this is related to the large temporal fluctuations in TC frequency and intensity, making it difficult to derive reliable

trends. Recent work has shown that, globally, a statistically significant trend towards an increase in TC intensity can be found





(Kossin et al., 2020). According to Knutson et al. (2010), future projections indicate an increase towards stronger storms of 2-11% by 2100, and a decrease in the globally averaged frequency of TCs by 6–34%, with a large variation between models and different basins. These general findings were confirmed in the modeling study by Knutson et al. (2015). The authors assessed, by means of CMIP5 model ensembles, the possible changes in TC frequency and intensity under RCP 4.5 for the late twenty-
first century compared to the period 1982-2005. Large differences between basins were depicted in the modeling study. For the North Indian Ocean (NIO) basin, an increase in TC frequency was estimated equal to 25.6% for TCs of category 1-5. The increase was even larger for TCs of category 4-5. An increase in intensification of TCs in the NIO during the last decades has already been reported by several authors (Webster, 2005; Singh et al., 2001; Singh, 2007; Deo et al., 2011; Kishtawal et al., 2012). According to Knutson et al. (2015), this increase in intensity was estimated to be 1.6% for TCs of category 1-5.
Nevertheless, the values described by different authors suggest a large scatter, making it difficult to derive statistically robust trends and conclusions. In combination with sea level rise, an increase in TC intensity will lead to significant increases in flood risk (Karim and Mimura, 2008).

In this study, TCWiSE was applied in combination with the hydrodynamic Delft3D Flexible Mesh model (Delft3D FM; 
Kernkamp et al., 2011) and the coupled wave model SWAN (Booij et al., 1999), to estimate storm surge and wave conditions along the BoB. TCWiSE was extended to be able to derive estimates for present and future scenarios, therefore accounting for the possible influence of climate change. In particular, downscaled projections of TC frequency and intensity based on Knutson et al. (2015) were used as input to the modeling study. Estimates of extreme storm surge and wave conditions were derived along the BoB (4,000+ km in total) at an alongshore resolution of 5-25 km following the coastal sections of the DIVA database 
(Hinkel and Klein, 2009). This data can be used as boundary conditions for the estimation of present and future hazards and risks resulting from TC events for the entire region and the conceptual design of suitable mitigation options, in combination with higher resolution local models. The methodology and tool are generic and can in principle be applied to any location worldwide that is exposed to TCs.

## 2 Data and methods

### 2.1 Study area

The BoB is the north-eastern part of the NIO, bounded on the southwest by Sri Lanka, on the west and northwest by India, on the north by Bangladesh and on the east by Myanmar. Within the bay lie the Andaman and Nicobar Islands (see 'ANI' in Figure 1). The different countries and regions considered in the study are described in Table 1. Per region, one representative location is included for further analysis and shown in Figure 1 ('Coastal cities'). The entire Bay of Bengal, and the northern 
part in particular, is highly affected by TCs. Extreme sea levels around the bay, including maximum tidal levels and storm



surge increase towards Bangladesh as a result of the bay geometry and the shallow continental shelf (Figure 1 and Muis et al., 2016).

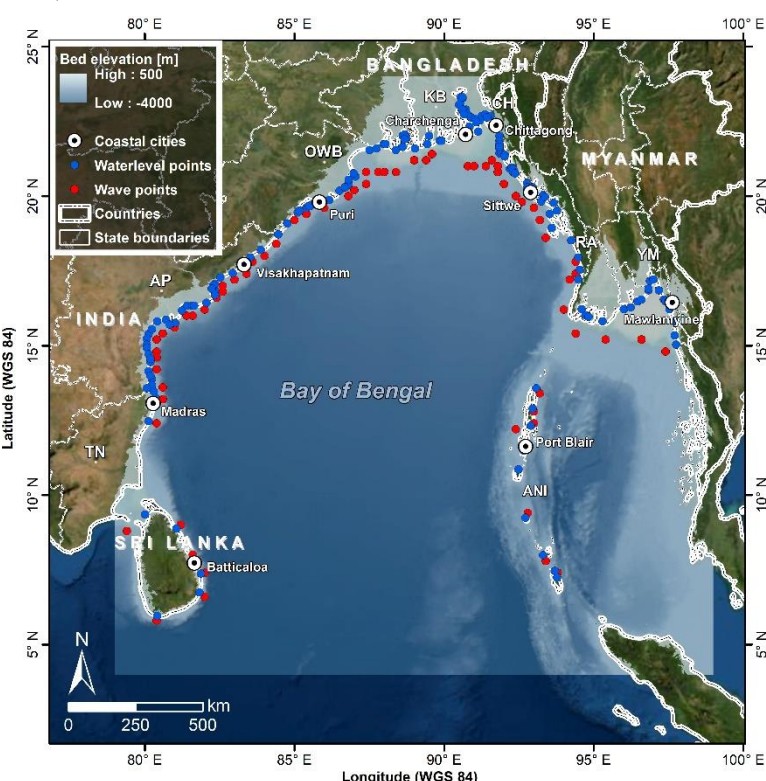

**Figure 1** Map of the Bay of Bengal with satellite image (ESRI) and bed elevation (GEBCO, Becker et al., 2009) including locations of
output points for detailed analysis at different coastal cities along the bay (white circles with dot), water levels (blue dots) and waves (red
dots). Country and state boundaries are indicated in white lines (ESRI). For a summary of countries, regions (with used abbreviations) and
locations for further detailed analysis, see Table 1.

**Table 1** List of countries, regions/states/provinces (abbreviation in brackets) and coastal cities for further detailed analyses. Charchenga is
sometimes also referred to as Charchanga (e.g. in Mamnun et al. (2020)).

| Country | Regions/States/Provinces | Coastal cities for detailed analysis |
|---|---|---|
| **Sri Lanka** | - | Batticaloa |
| **India** | Tamil Nadu (TN) | Madras |
| | Andhra Pradesh (AP) | Visakhapatnam |
| | Odisha and West Bengal (OWB) | Puri |
| | Andaman and Nicobar Islands (ANI) | Port Blair |
| **Bangladesh** | Khulna and Barisal (KB) | Charchenga |
| | Chittagong (CH) | Chittagong |
| **Myanmar** | Rakhaing and Ayeyarwady (RA) | Sittwe |
| | Yangon and Mon (YM) | Mawlamyine |





## 2.2 Data

Different datasets were used in order to set-up the numerical modeling system, namely: bathymetry, coastal segments and HTC tracks with associated maximum wind speeds.

Deep-water bathymetry data for the BoB were derived from the GEBCO 2008 global bathymetric data set (Becker et al., 2009), see Figure 1). These bathymetric data were used as input for both the hydrodynamic and wave models.

Coastal segmentation from the DIVA schematization (Hinkel and Klein, 2009) was used as a basis to define the locations at which to extract time-series of extreme storm surge and significant wave heights along the entire coastline, including 197

points in total. Each segment has a length of approximately 5-25 km. For the wave conditions, these DIVA segments were translated into locations in deeper water to provide deep-water wave conditions, therefore not affected by the local bathymetry. For each location, the closest point, with a water depth larger than 30 meters, was chosen (Figure 1).

HTC data were derived from the IBTrACS (International Best Track Archive for Climate Stewardship) database version

v04r00 (Knapp et al., 2018). The IBTrACS database contains the most complete global set of HTCs available. From the IBTrACS dataset, the best track data of the Joint Typhoon Warning Center (JTWC) for the NIO was chosen as source, including reliable satellite-derived data (Singh, 2010) available for the period 1972-2020. The data contains TC information including the best track coordinates and maximum wind speeds. The 1-minute averaged wind speeds were converted into 10-minute wind speeds using, as correction factor, the value 0.93, following Harper et al. (2010). In total, 110 historical tracks

were available for the NIO, of which 81 originated in the BoB including the recent cyclone Amphan (2020). In the dataset, two distinct periods with TC activity can be identified, corresponding with the pre-monsoon period (May) and the post-monsoon period (November, e.g. Alam et al. (2003) and Islam and Peterson (2009)). Generally, about 2-4 TCs per year are generated in the NIO, though this is not spread evenly through time, as the TC generation is influenced by a number of external factors such as the El Niño-Southern Oscillation (ENSO) cycle (e.g. Singh et al. (2000), Hoarau et al. (2012)). The HTC tracks

were used to derive the STC tracks in TCWiSE. Figure 2a illustrates the selected HTC tracks, with different colors indicating different wind speed categories, while Figure 2b shows the generated STC tracks based on the HTC. Note that wind speeds on land may be less reliable as they are affected by several factors (e.g. land friction, local topography). Nevertheless, these uncertainties are not relevant for this study, which instead focuses on storm surge and waves in the ocean and coastal zone.

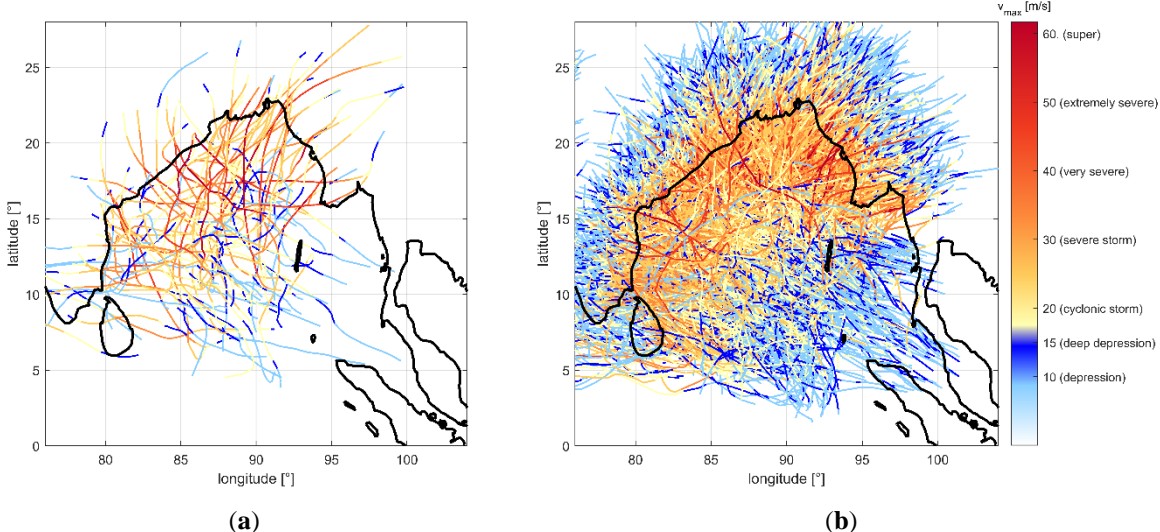

(**a**)                                            (**b**)

**Figure 2** Cyclone tracks subdivided in different cyclone wind speed categories based on the intensity scale of the India Meteorological
Department: (**a**) Historical Tropical Cyclones (HTC) for the period 1972-2020; (**b**) Synthetic Tropical Cyclones (STC) for a period of 1,000
years.

## 2.3 Methods

The methodology which was followed for generating extreme values for surge and waves for both historical and synthetic

tracks is described in Figure 3. From the IBTrACS dataset, a regional subset of HTC is extracted for the NIO. In particular,

TC characteristics (i.e. location, wind speed) describing the HTC are extracted and used for generating the STC with TCWISE.

The HTC and STC are then  converted into wind and pressure fields using WES (Wind Enhancement Scheme; Deltares, 2019),

which is a routine incorporated within TCWiSE. The generated wind and pressure fields are then used to force the numerical

hydrodynamic and wave models Delft3D FM and SWAN. From these models, time-series of storm surge and wave heights

are extracted at the DIVA segments and, per location, an Extreme Value Analysis (EVA) is performed using the peaks-over-

threshold (POT) method. The results are extreme values of storm surge and wave heights for different return periods at each

location, which are used to perform regional comparisons between the obtained HTC and STC datasets.

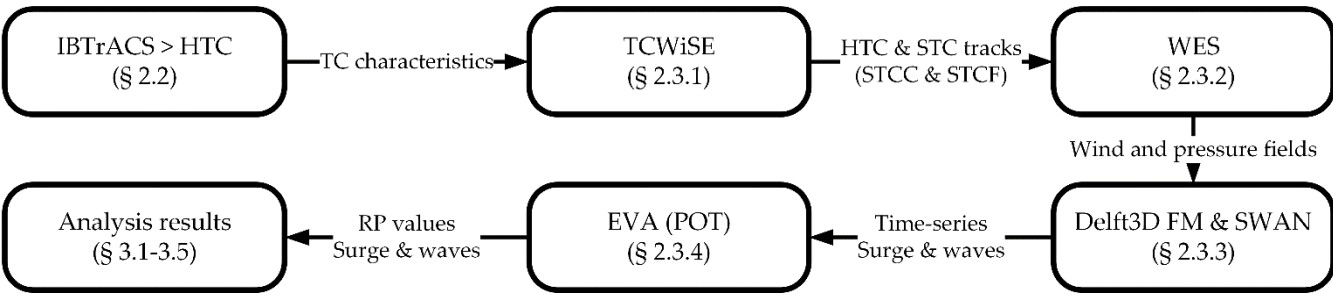

**Figure 3** Flow diagram showing the procedure to generate regional comparisons of surge and waves values. Abbreviations: IBTrACS
(International Best Track Archive for Climate Stewardship); HTC (Historical Tropical Cyclone); TCWiSE (Tropical Cyclone Wind



Statistical Estimation tool); HTC (Historical Tropical Cyclone); STC (Synthetic Tropical Cyclones); STCC (Synthetic Tropical Cyclones Current climate); STCF (Synthetic Tropical Cyclones Future climate); WES (Wind Enhancement Scheme); POT (peaks-over-threshold method); EVA (Extreme Value Analysis); RP (Return Period). The section numbers at which each of the steps is elaborated is also indicated.

### 2.3.1 Generation of synthetic cyclone tracks

The generation of STC tracks was carried out using TCWiSE (Nederhoff et al., 2021). The tool allows the generation of synthetic tracks based on a Markov model where observed data serves as a data source to compute synthetic tracks. The main variables it keeps track off are location (latitude and longitude), time and the statistics of maximum sustained wind speeds ($v_{max}$), forward speed (c) and heading (θ) as spatially-varying PDFs (Probability Density Functions). TC genesis is computed through randomly sampling the locations for each track from a spatially-varying PDF. TC termination is estimated based on

PDFs describing the probability that a TC will terminate at a certain location and with a given wind speed. These spatially-varying PDFs are all constructed based on historical input data and created on a 0.1-degree grid for the entire NIO. Furthermore, the number of TCs per year and the probable period within the year of TC generation are used to provide each track with a unique time within the synthetic year.

At first, the 81 HTC tracks which have occurred in the BoB over a period of 48 years, with location and $v_{max}$, were extracted from the IBTrACS dataset (Figure 4). After calculating the PDFs of the different variables based on the HTC, TCWiSE was run to estimate 1,000 years of synthetic tracks both for current (Synthetic Tropical Cyclones Current climate, STCC) and future climates (Synthetic Tropical Cyclones Future climate, STCF). The estimation of the effects of future climate on TC was based on Knutson et al. (2015). The authors estimated that the frequency of TCs per year may increase with 25.6% by the end of the

century, for all TC categories (cat 1 – 5), defined as TC with wind speeds larger than 33 m/s, and according to an RCP 4.5 scenario. Similarly, the intensity (i.e. maximum wind speed) may increase by 1.6%. In order to avoid creating sampling differences when creating the synthetic tracks for current and future scenarios separately, and making sure that only frequency and intensity would vary, the STCF were generated first, resulting in 2191 tracks. Then, the first 1745 tracks were selected as representative of 1000 years TC in the current climate (Figure 4). The ratio between these two values (2191/1745) represents

the 25.6% TC frequency increase between future and current scenario. Similarly, the wind speeds for all time steps of the STCF was reduced by 1.6% to create the wind fields of the STCC.

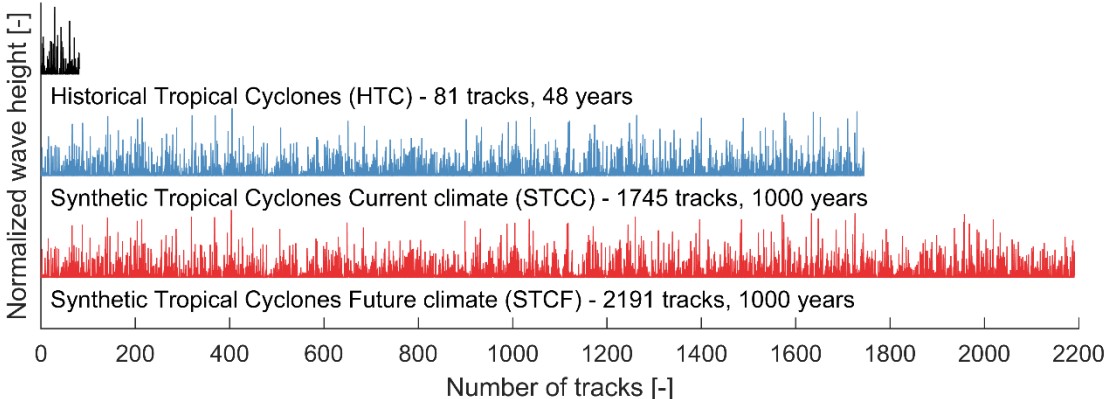

**Figure 4** Normalized wave height and number of tracks for HTC (black), STCC (blue) and STCF (red) at Charchenga, Bangladesh.

## 2.3.2 Generation of pressure and wind fields

For each HTC and STC track, time- and spatially-varying wind and pressure fields were generated based on the parametric wind model of Holland et al. (2010) using WES. The wind-pressure relations of Holland (2008) were used to compute the maximum pressure drop and create corresponding spatially-varying pressure fields. The calibrated coefficients for the NIO

basin based on Nederhoff et al. (2019) were used to compute TC geometry (most probable Radius of Maximum Winds (RMW), and radius of gale force winds (R35)). TC asymmetry between the different quadrants was defined using Schwerdt et al. (1979). Furthermore, a constant inflow angle of 22 degrees based on Zhang and Uhlhorn (2012) was assumed and a wind decay after landfall was included based on Kaplan and DeMaria (1995).

## 2.3.3 Hydrodynamic and wave numerical models

The effect of each individual TC on storm surge and wave conditions were modelled using a coupled hydrodynamic (Delft3D FM) and wave model (SWAN). The resolution of the models was chosen in such a way to keep the running time sufficiently low to be able to run the entire set of thousands of tracks, and still allowing sufficient accuracy to simulate realistic results. For the hydrodynamic model, the finest grid size was equal to about 3 km nearshore and extracted from the GTSM (Global Tide and Surge Model; Muis et al., 2016) grid and forced with wind and pressure fields only from the TCs. Tidal forcing was

excluded from the model to be able to intercompare the impact of different TCs directly and focus on the wind- and atmospheric pressure-driven surge components only, without the timing of the actual high- or low-water of the tide playing a role. In reality, non-linearities between tide and surge can, depending on the tidal range, have a large effect on the estimated total water level values (Chiu and Small, 2016).





For the wave model, a constant resolution of 0.02 degrees (~2 km) was used to allow the simulation of the large number of tracks. This resolution is not sufficient to accurately model nearshore processes like wave shoaling and breaking. Therefore, only results in water depths deeper than 30 meters were used. The wave model was run in non-stationary mode, with a time step of 10 min, and forced by wind fields from the TCs only. No background winds or wave heights were included since these are not available for the synthetic tracks and they would alter the comparison with the historical tracks. For more details
regarding the setup of the numerical models see Appendix A. Per DIVA section, the generated time-series of storm surge and significant wave heights per track were combined to generate 1000-year long time-series of STCs. Additionally, also a 48-year long time-series of storm surge and wave heights was created forcing the models with the HTC winds and pressures.

**2.3.4 Extreme value analysis**

An EVA was carried out on the extreme storm surge and wave heights using a POT-method and an exponential fit (Caires,
2016). Extreme values were derived along the entire coastline at each DIVA-segment and based on the created time-series, for return periods corresponding to 10, 25, 50 and 100 years. The POT thresholds were automatically selected between the 99-percentile, as minimum threshold, and the 99.9-percentile as maximum threshold, using the threshold stability criteria (Caires, 2016). For the HTC, the 98-percentile was used as a minimum percentile to create stable EVA fits as the time-series contained a lower number of peaks. Additionally, 95% confidence intervals (CIs) were computed and based on the CI of a Generalized
Pareto Distribution (GPD), by applying the relative values of the lower and upper estimates, relative to the point estimates. The point estimates are determined with as plotting position (xi, (n+1)/(lu (n+1-i)), where n is the sample size, i the order and lu the Poisson rate). Hereafter, for the comparison between the HTC/STCC/STCF results, the absolute values per DIVA section were averaged into larger regions, as described in Table 1.

**3 Results**

**3.1 Verification synthetic cyclone tracks**

Time-series of 1,000 years of synthetic tracks for the BoB for current (STCC) and future climate (STCF) were generated using TCWiSE. Since the synthetic tracks were based on historical data, statistical properties for the STCC should be similar to those of the HTC. Nederhoff et al. (2021) demonstrated this through an application of TCWISE for the Gulf of Mexico, by comparing statistical properties for 10,000 years of synthetic tracks and those of 133 years of historical data.


Here, a similar analysis is presented for the BoB. Spatial patterns for genesis, occurrence and termination were compared qualitatively and quantitively using the correlation parameter R based on the similarity score of Kirchhofer (1974) in Figure B1 in the Appendix. The closer R is to 1, the more similar the spatial patterns of the HTC and subsequently generated STCC are. The regions with the highest probability of TC genesis in the historical data were found around the Andaman and Nicobar





Islands in the middle of the BoB (Figure B1a and B1b). The spatial genesis patterns of the STCC appear very similar to those of the HTC, which is confirmed by the correlation parameter R being 0.88 [-]. For the probability of termination of TCs, the patterns appear similar as well (Figure B1c and B1d), but the magnitudes are more spread out for the STCC leading to a slightly lower R-value of 0.82. The main regions of TCs making landfall is around Eastern India, Bangladesh and Northern Myanmar. The highest TC occurrence is found in the middle of the BoB (Figure B1e and B1f). The yearly occurrences are quite well

reproduced with an R-value of 0.68. The lower correlation is due to the spatial patterns of the synthetic tracks being more smoothed out compared to that of the HTC due to the much larger number of realizations. The maximum yearly probability of the HTC is about 0.4 (i.e. return period of 2.5 years), indicating that a particular region in the NIO is likely to be affected by a TC in average once every 2.5 years. The spatial coverage of the probability of genesis, probability of termination and yearly probability estimation over the whole BoB, as well as the similarities in spatial patterns between the HTC and the STCC,

indicate that the STCC can be used as a basis for quantifying wave heights and storm surges along the entire coast.

A comparison between the Cumulative Distribution Functions (CDF) of maximum wind speeds ($v_{max}$) for HTC and STCC, at nine locations along the BoB, is shown in Figure 5. The functions are estimated based on TCs within a 200 km radius from each of the nine locations. Per location, the number of samples of timesteps of TCs within a 200 km radius are included; these

are between a factor 6 and 34 larger for the STCC compared to HTC and including 1000 samples or more except for Mawlamyine (Myanmar). The CDFs of the other parameters (i.e. forward speed and heading) are presented respectively in Figure B2 and B3 of Appendix B. Additionally, a number of statistical parameters describing the differences between the two distributions are computed (i.e. absolute difference in maxima (Δmax), normalized Mean Absolute Error (nMAE), the relative bias of the median value (bias), and the Root-Mean-Square Error (RMSE) of the whole CDF function). Locations characterized

by a higher HTC occurrence, as the ones in India and Bangladesh, generally have a lower absolute bias. For locations with a lower HTC occurrence, as for example Batticaloa (Sri Lanka) and Mawlamyne (Myanmar), the bias is larger and the CDFs of the HTC indicate a less smooth distribution as a result of the lower number of tracks included. The nMAE varies between 0.01 and 0.06, according to the location, and the RMSE between 1 and 10 m/s, with the largest discrepancies seen at Batticaloa. At Batticaloa, there are a limited number of samples for the HTC with a clear distinction of tracks with a wind speed of either

below 15 or above 25 m/s (Figure 5), while for the STCC a more gradual distribution can be seen, including also realizations in between these values. Maximum wind speeds are generally higher for the STCC than for the HTC, meaning that more extremes are captured in the STCC, and with largest differences observed at Port Blair reaching up to 25 m/s. Smoother distributions of the parameters describing the TCs is one of the advantages of using synthetic tracks computed over a long period of 1,000 years. Similar patterns are also visible for the forward speed and heading (Figure B2 and B3). In these cases,

the RMSE for the forward speed ranges between 0.4 and 1.4 m/s (nMAE 0.1-0.5) while the RMSE for the heading range between 11.3 and 50.5 degrees (nMAE 0 – 0.04).

Based on these results it can be concluded that the first-order hazards of wind speeds are well resembled by the synthetic tracks created by TCWiSE for the current climate compared to historical observations. Therefore, the computed synthetic tracks and

wind speed will be used for further analysis in the paper to estimate the resulting second order hazards (i.e. storm surge and wave height).

**Figure 5** Comparison between CDFs of maximum wind speed ($v_{max}$) for HTC (black line) with 75% confidence intervals (dashed line) and STCC (blue line) at nine locations along the Bay of Bengal. The functions are estimated based on TCs within a 200 km radius from each

location. The number of samples within the 200 km radius is indicated (#HTC and #STCC), alongside several statistical parameters comparing the HTC and STCC distributions (i.e. absolute difference in maxima (Δmax), normalized Mean Absolute Error (nMAE), the relative bias of the median value (bias), and the Root Mean Square Error (RMSE) of the whole CDF function. The nine locations are shown in Figure 1.



## 3.2 Convergence of synthetic results

After validation of the synthetic cyclone tracks, spatially varying wind fields were created and used as input to force coupled Delft3D FM and SWAN models to estimate storm surge and wave heights along the BoB. It is first verified how dependent the estimated variables of wind speed, wave heights and storm surge are to the number of years of synthetic data included and if/how fast the results converge. In Figure 6 an example for Charchenga (Bangladesh) is shown. To verify the convergence, 99,000 EVAs were performed on the time-series of wind speeds, wave heights and storm surge to estimate the 100-year return

period value, while increasing the "X" numbers of years extracted from the available 1000-year time-series (i.e. X = 10, 20, 30, …, 1000 years). Hereby for every "X" numbers of years extracted, these years are selected randomly and combined into 1 new time-series after which an EVA is computed. This is repeated 1,000 times per "X" numbers of years included to get a stable estimate. Estimates of the 2.5$^{th}$, 50$^{th}$ and 97.5$^{th}$ percentiles for a 100-year return period value were computed at each iteration. Therefore, per X years of data included, the 95% CI spread of the estimated median value was calculated over the

1000 realizations, which is shown in Figures 6a,c,e as green fill. The estimated median value of this spread is included as orange line. The same was calculated for the 2.5$^{th}$ and 97.5$^{th}$ percentile values, with their respective median values of the spread shown as grey and black lines respectively and with their combined total spread as gray fill. Figures 6a,c,e show that the more years of synthetic data are included, the smaller the CIs become and the more the median values converge to a stable value. The number of years of data to be included to reach this stable value depends on the variable that is analyzed.


To quantify how quickly each of the different variables converge, the relative (%) ratio between the median value of the 50$^{th}$ percentile (i.e. computed based on X years of data) and the same median value based on 1000 years of data (i.e. here assumed to be the "true value"), is presented as orange lines in Figures 6b,d, f. These results show that, for all variables, the convergence is exponential. While the variables wind speed and wave heights already have a relative difference of less than 5% (orange

dashed line) compared to the "true value" after about 380 and 350 years respectively, for the storm surge this takes about 450 years. On the same figure, the range describing the difference in spreading of the median value (i.e. the size of the green fill in panels 6a, 6c, 6e) is also shown as green line. This range indicates how 'wrong' the estimated median values can be after including a certain number of years of data. After including 200 years of data, the 95% spread of the median values is still 15 m/s for the wind speed, 5 m for the wave height and 0.8 m for the storm surge. This reduces more rapidly for the wave heights,

and more slowly for the wind speed and storm surge. The difference in convergence can be explained by the related probability distribution for each of the variables, also known as 'tail dependence' in probability theory. For wind speed and storm surges the range of possible values is generally larger (type I tail = no upper limit), while this is more limited for wave heights (type III tail = with upper limit). This can be explained physically by the influence area of variables wind speed and resulting storm surge that is limited to close to the landfall location of a TC. Swell waves generated by a TC can travel hundreds of kilometers

and have a larger area of influence, so that higher wave heights are more often reached, and the range of possible values is more limited. Since all 3 variables reach a relative difference of maximum 5% within 450 years and a maximum difference of




2% (orange dotted line) within using 830 years of data, we can assume that results have converged after including the full 1000 years of data and these are further used in this analysis for comparison with the HTC data.

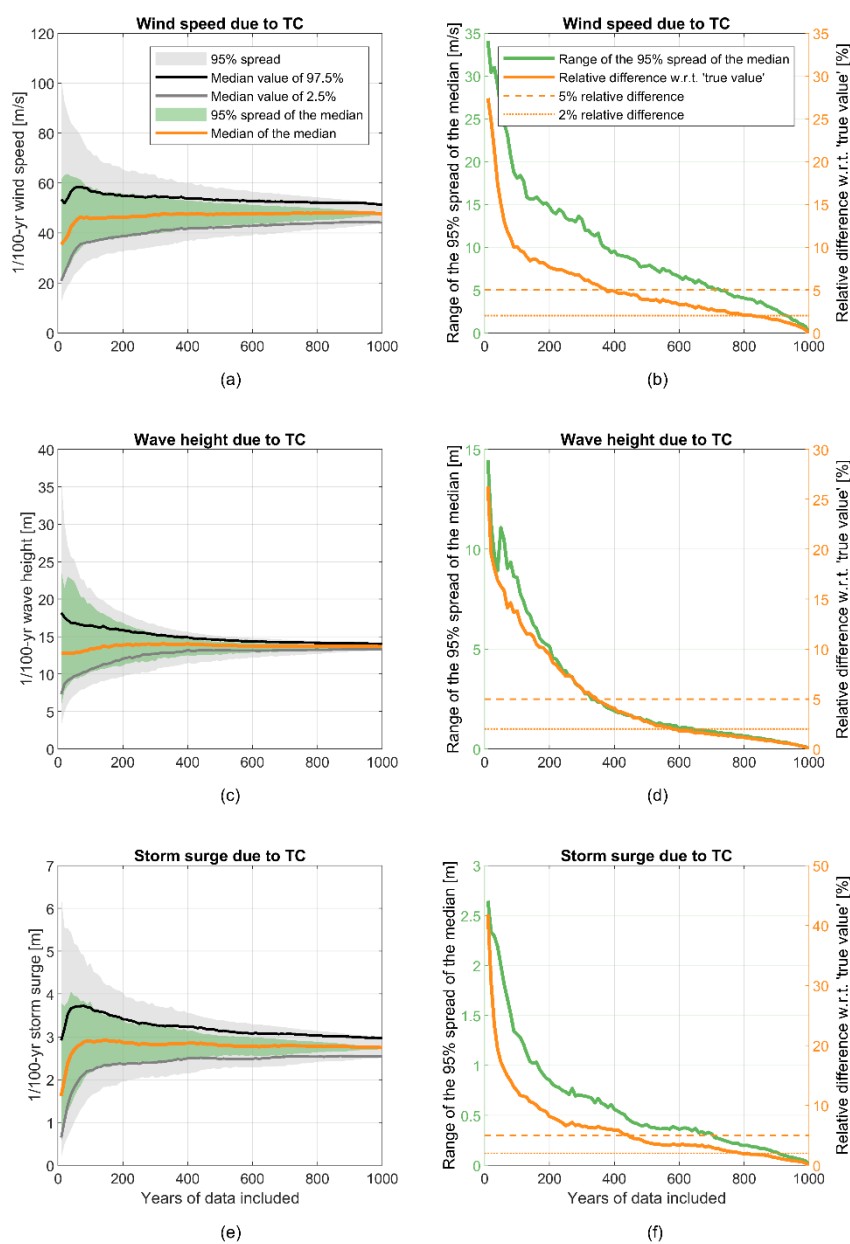





**Figure 6** Verification of the convergence of the 100-year return period computed wind speed, wave height and storm surge as a function of the number of years of STCC tracks included in the analysis. Panels (a), (c), (e), respectively refer to wind speed, wave height and storm surge. The 95% spread (97.5th - 2.5th percentile) is indicated by the grey fill, the median value of the 97.5th percentile is shown as black line and the median value of the 2.5th percentile is shown as grey line. The green fill shows the 95% spread of the median values, while the orange line indicates the median of the median value. Panel (b), (d), (f) show the range of the 95% spread of the median value (green line) and the relative difference with respect to the 'true value', being the estimate based on 1,000 years of STCC tracks (orange line), respectively for wind speed, wave height and storm surge.

### 3.3 Evaluation of predicted storm surge and wave heights for historical versus synthetic tracks

### 3.3.1 Local differences based on one location

To compare the results of predicted storm surge and wave heights based on historical and synthetic tracks for the current climate, first an EVA is performed and discussed for one example location at Charchenga. Afterwards, results for all points combined are presented as general differences based on all modelled locations (Section 3.3.2), as well as differences between regions within the BoB (Section 3.3.3). In all figures, results for HTC are shown in black, for STCC in blue and for STCF in red.

Focusing at the differences at one location first, Figure 7 shows a comparison of the EVA specifically for storm surge at Charchenga. The EVA for the HTC is based on 14 data points only, with just 4 points above a return period of 10 years. Given the length of the record equals 48 years, the maximum value of the dataset has an estimated return period of approximately 48 years. Thus, in order to obtain values for longer return periods (e.g. 100-year return period) one needs to extrapolate from the fitted GPD. For the STCC, there are 100 data points that have a return period of at least 10 years and the maximum return period in the dataset is approximately 1000 years. This leads to a much smaller 95% CI for the STCC. In particular, the STCC has a 95% CI of 0.95 m for a return period of 100 years versus 2.34 m for the HTC.

For return periods lower than 5 years the point estimates of the HTC and STCC convergence. This gives further confidence in the local validity of the STCC estimates, in addition to the verification as described in sections 3.1 and 3.2. The estimated storm surge for the 100-year return period are underestimated when using the HTC with respect to the use of the STCC. This underestimation is a result of a limited number of data points on which to fit the GPD (i.e. under sampling). This may lead to a different fit, as shown in Figure 7, where the lines of the HTC and STCC are approximately parallel but that of the STCC is shifted upwards (Figure 7). The result is, for example, that the highest value modelled of the HTC, with a storm surge close to 2 meters, gets a return period estimate close to the length of the dataset of 48 years. However, based on the STCC results, this event has a return period of only 30 years.

In the STCC the maximum modelled surge is 1.5 m higher than observed with historical events. Given that the CDF of the maximum wind speed for HTC and STCC are very similar and the maximum wind speed is only 4 m/s larger (Figure 5), this means that more disadvantageous (but statistically plausible) trajectories in terms of land fall location, heading and forward





speed are included in the synthetic dataset compared to historic events, meaning that the worst event for this location may not have happened yet in recorded history. The same figure, but for wave heights, is shown in the appendix (Figure B4).

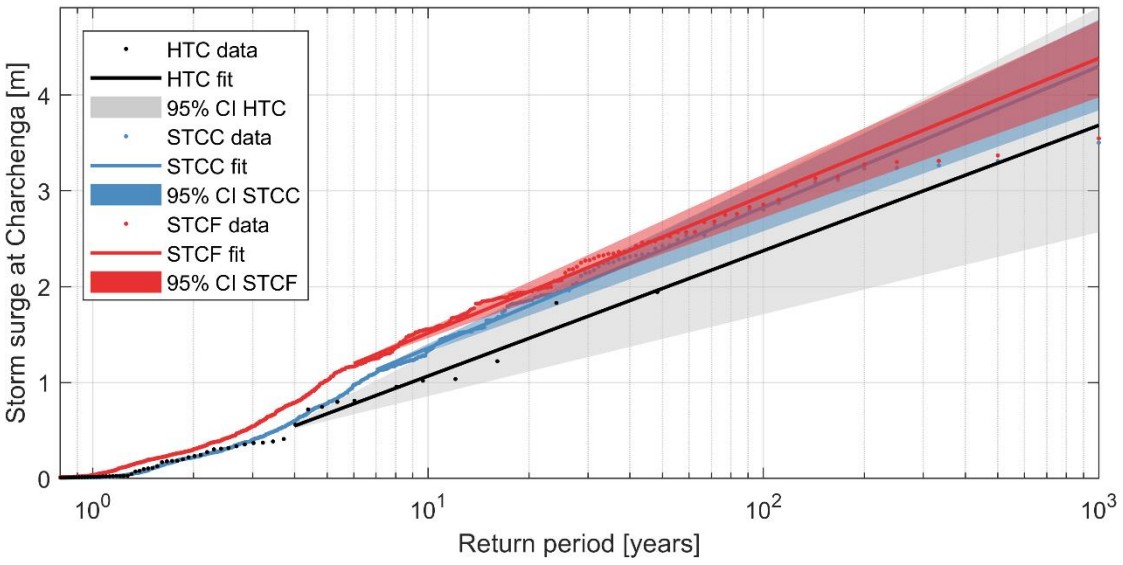

**Figure 7** Example of an EVA for storm surge at Charchenga, Bangladesh, for HTC (black), STCC (blue) and STCF (red). The horizontal axis represents the return period in logarithmic scale, while the vertical axis represents the storm surge in meters. Shown are the data points with respective return periods (dots), the EVA fit (solid line) and the 95% confidence intervals (colored fills).

### 3.3.2 General differences based on all locations

Besides analyzing the difference in modelled values and CIs of the EVA at a specific location (Figure 7), this was also done

for all 197 DIVA segments together, over the entire BoB. Figure 8 shows the results presented as scatter plots for different return periods both for storm surge (panel a) and wave height (panel b), where computed values based on HTC are shown on the x-axis and computed values based on STCC are shown on the y-axis. The figure shows that, with increasing return period, the Root-Mean-Square Difference (RMSD) also increases. Computed storm surges are in general slightly larger when using the STCC (increase of ~5 – 10% depending on the return period), while computed wave heights are in general larger when

using the HTC (increase up to ~5% for larger return periods, though with more scatter). These increases for the HTC/STCC respectively are calculated as relative bias with respect to HTC, though since the values based on HTC are not the 'true values', this is referred to as a trend. In Figure 9, the same information is shown but for the 95% CI, computed as the difference between the 97.5[th] and the 2.5[th] percentile estimates. The 95% CI are significantly smaller for the STCC compared to the HTC with a trend between 60 and 80%.




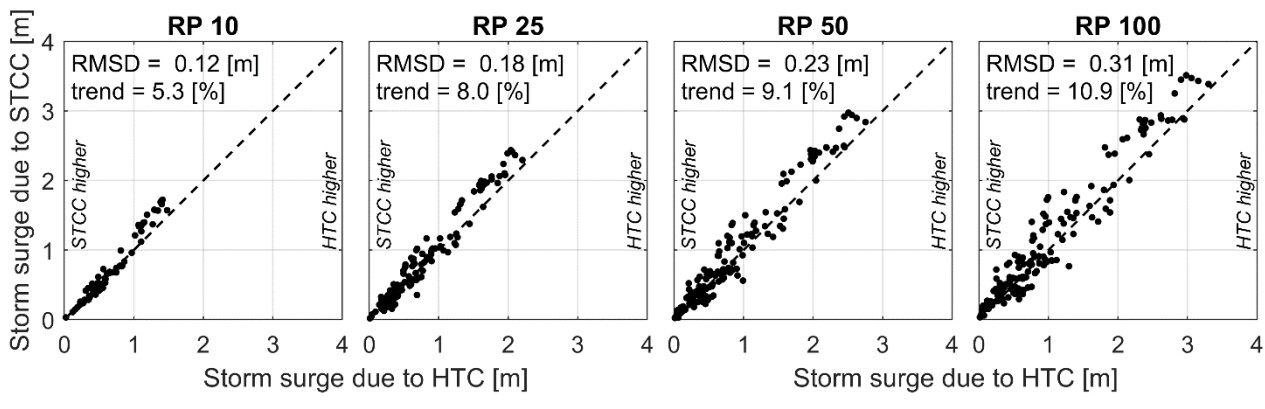

**(a)**

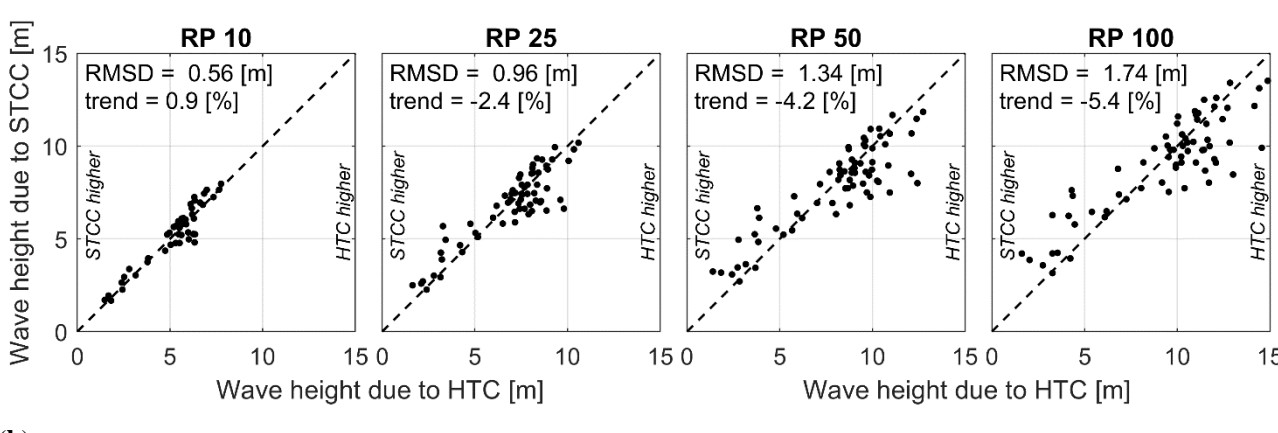

**(b)**

**Figure 8** Scatterplots of computed storm surge (panel a) and wave heights (panel b), both resulting from HTC (x-axis) and STCC (y-axis), for return periods of 10, 25, 50 and 100 years, for all locations along the Bay of Bengal. Root-mean-square differences (RMSD) and trend (%) of STCC compared to HTC are also shown.

Natural Hazards
and Earth System

**(a)**

**(b)**

**Figure 9** Scatterplots of 95% confidence intervals (97.5th minus 2.5th percentile) for storm surge (panel a) and wave heights (panel b), both resulting from HTC (x-axis) and STCC (y-axis), for return periods of 10, 25, 50 and 100 years, for all locations along the Bay of Bengal. Root-mean-square differences (RMSD) and trend (%) of STCC compared to HTC are also shown.

### 3.3.3 Regional differences within the Bay of Bengal

The differences between storm surge and wave heights computed based on HTC and STCC also vary spatially within the BoB. At locations where TCs occur more frequently, as for example Bangladesh, patterns as described in Section 3.3.1, for the predicted storm surge at Charchenga, can be observed (Figure 10). At locations with a lower TC occurrence, as for instance Batticaloa (Sri Lanka), the use of HTC to estimate storm surge leads to much larger confidence intervals and lower values of the storm surge than the one predicted based on STCC, which could be an underestimation of the potential hazard using the HTC. Since the number of TCs making impact at these locations is very limited, TCs that could possibly hit these stretches of coast may not have occurred yet in the historical events. Therefore, using a large synthetic dataset largely reduces the confidence intervals and improves the estimation of the coastal hazards of extreme events since a larger range of possible





events is covered. For wave heights, the differences between using HTC and STCC are smaller (Figure 11). This is because waves are in general a less local effect than storm surge with extreme waves at one location possibly being the result of a TC passing at a certain distance from a specific location, leading to more events for the HTC. The estimated values for waves based on STCC also fall within the CI of the HTC estimates, while for storm surge this is not always the case.


To visualize these regional differences, the estimated regionally-averaged values for storm surge and wave heights along the BoB based on the STCC are presented respectively in Figure 12a and Figure 13a as well as summarized in Table 2 of section 3.5. The regions with higher values for storm surge and wave heights are also the regions with a higher TC occurrence. Furthermore, the presence of a wide shallow continental shelf as in front of Bangladesh contributes to further amplifying the

storm surge as a result of a larger wind-driven setup. Therefore, the highest storm surge can be found there. Over the entire BoB, the average storm surge due to TCs is estimated to be 1.2 m for a 100-year return period event. For waves (Figure 13a) there is relatively less regional variability because wave impact due to TC is a less local event than storm surge, though still some differences are visible. The largest wave heights are again found in front of the coast of Bangladesh. The averaged deep-water significant wave heights over the entire BoB is 9.5 m for a 100-year return period.


The relative differences in estimated storm surge and wave heights computed as HTC compared to STCC are shown respectively in Figure 12b and Figure 13b. The largest differences are observed for the storm surge, and in particular in the southern part of the BoB, with differences of more than 50%. The storm surge is relatively small here and can be both relatively larger/smaller for storm surge derived based on STCC than those derived based on HTC (see also Table 2). In the northern

part of the BoB (i.e. Bangladesh) differences are relatively smaller (less than 20%), however the difference in magnitude is much larger (see also Table 2). Storm surge estimated based on STCC are here consistently larger than those derived based on HTC for all return periods.

For the wave heights the differences are in general smaller. In the south of the BoB (Sri Lanka, Andaman and Nicobar Islands),

the use of HTC leads to an underestimation of the predicted wave height with respect to the use of STCC for larger return periods, while along the main continent an opposite behavior can be seen, although with relatively minor differences (see also Table 2).

Comparing the sizes of the CI per region as percentage of the computed absolute value based on STCC (of Figure 12a and

Figure 13a), the size of the CI for storm surge is much smaller based on the STCC (Figure 12c) as opposed to based on the HTC (Figure 12d). The maximum size for the STCC is 35% compared to the computed 100-year return period value, while for the HTC the CI can be just as large as the computed value or higher (>100%). The same holds for the wave heights (Figure 13c), which is with 25% for the STCC much smaller than the 80% based on HTC (Figure 13d).


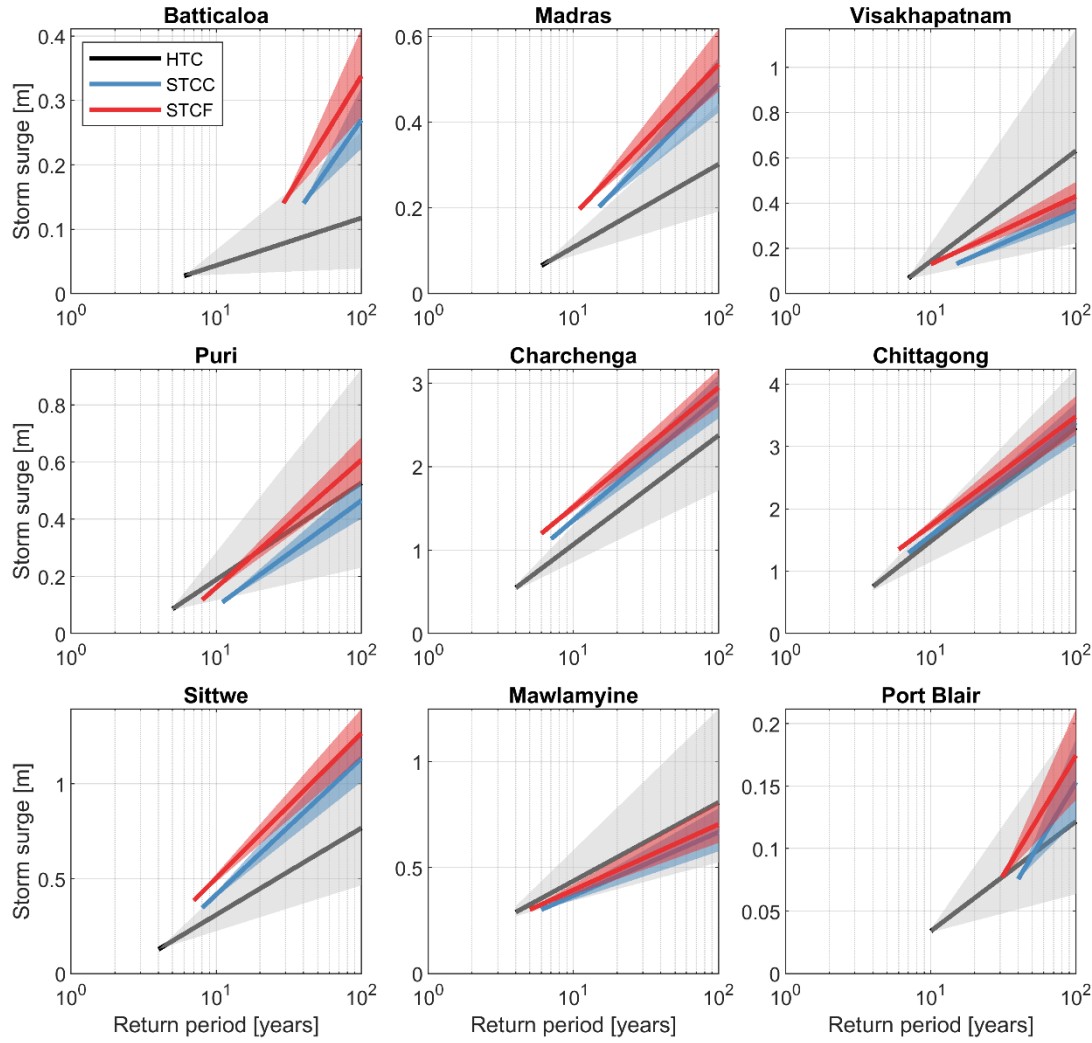

**Figure 10** Extreme value analysis for 9 different locations for storm surge based on HTC (black), STCC (blue) and STCF (red). Per panel both the fit (solid line) and 95% confidence intervals (background fill) are included. The horizontal axis is return period in logarithmic scale, the vertical storm surge in meters. Note: orientation of locations goes in clockwise direction through the Bay of Bengal from Batticaloa to Port Blair and y-axis varies per location.



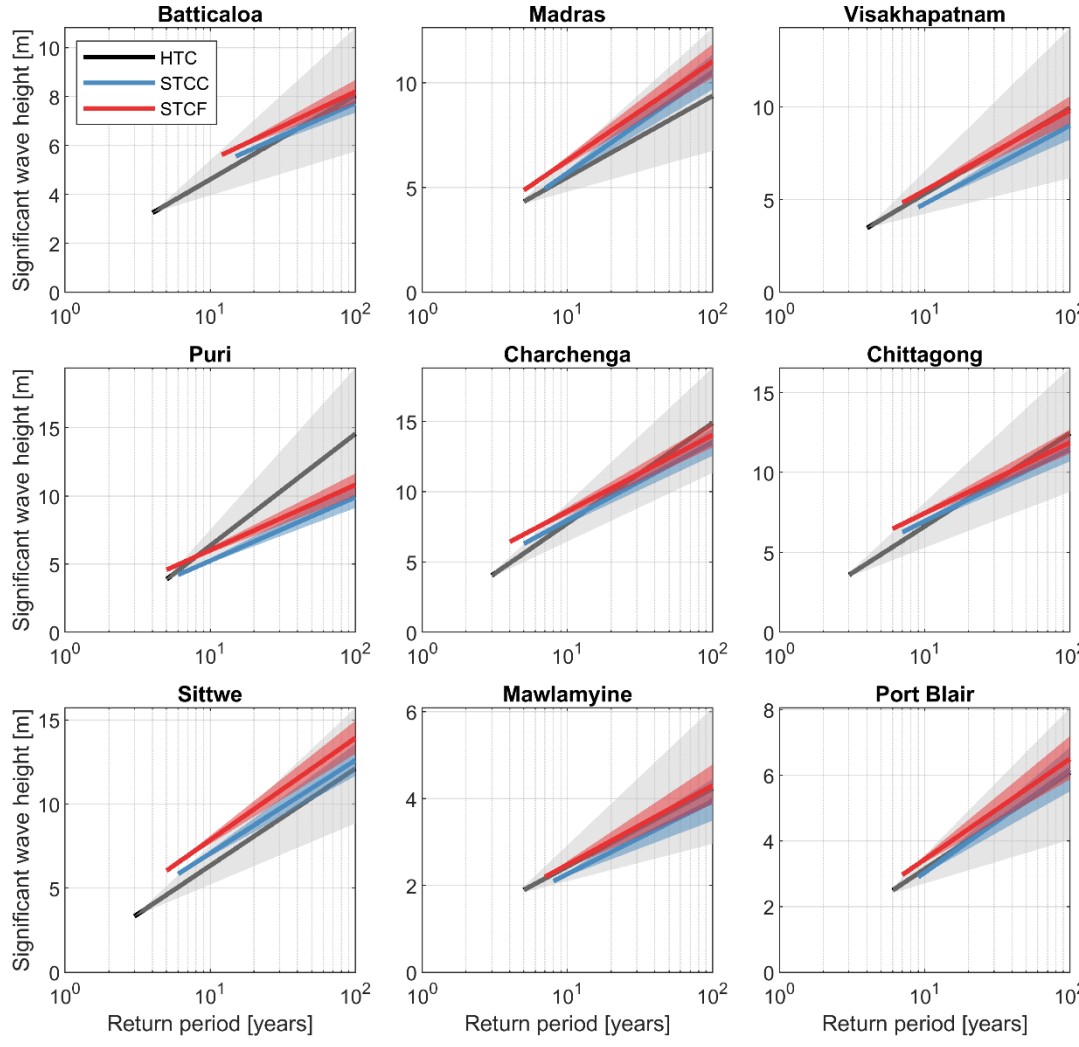

**Figure 11** Extreme value analysis for 9 different locations for significant wave height based on HTC (black), STCC (blue) and STCF (red). Per panel both the fit (solid line) and 95% confidence intervals (background fill) are included. The horizontal axis is return period in logarithmic scale, the vertical significant wave height in meters. Note: orientation of locations goes in clockwise direction through the Bay of Bengal from Batticaloa to Port Blair and y-axis varies per location.




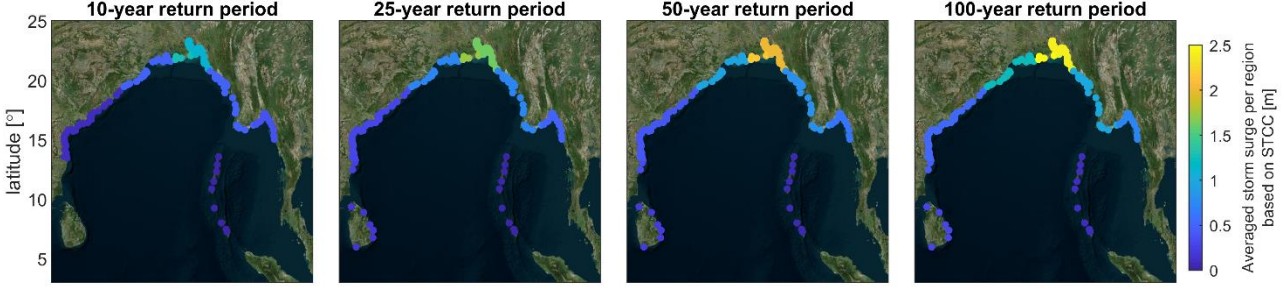

(a)

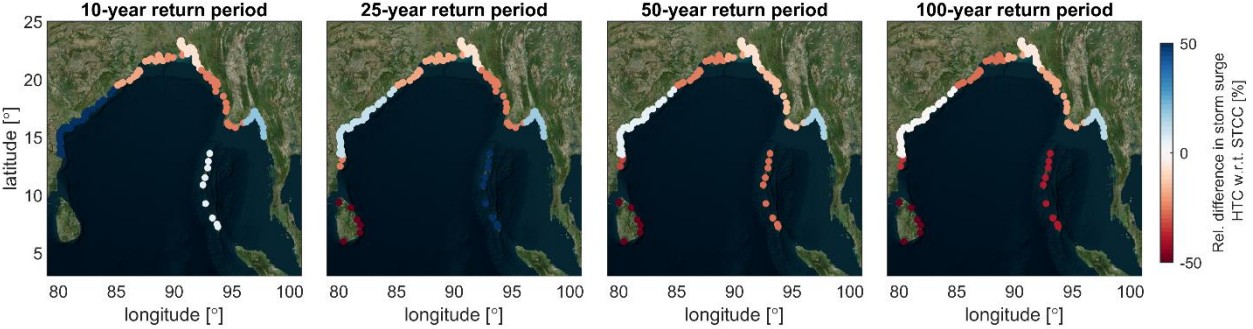




**(b)**

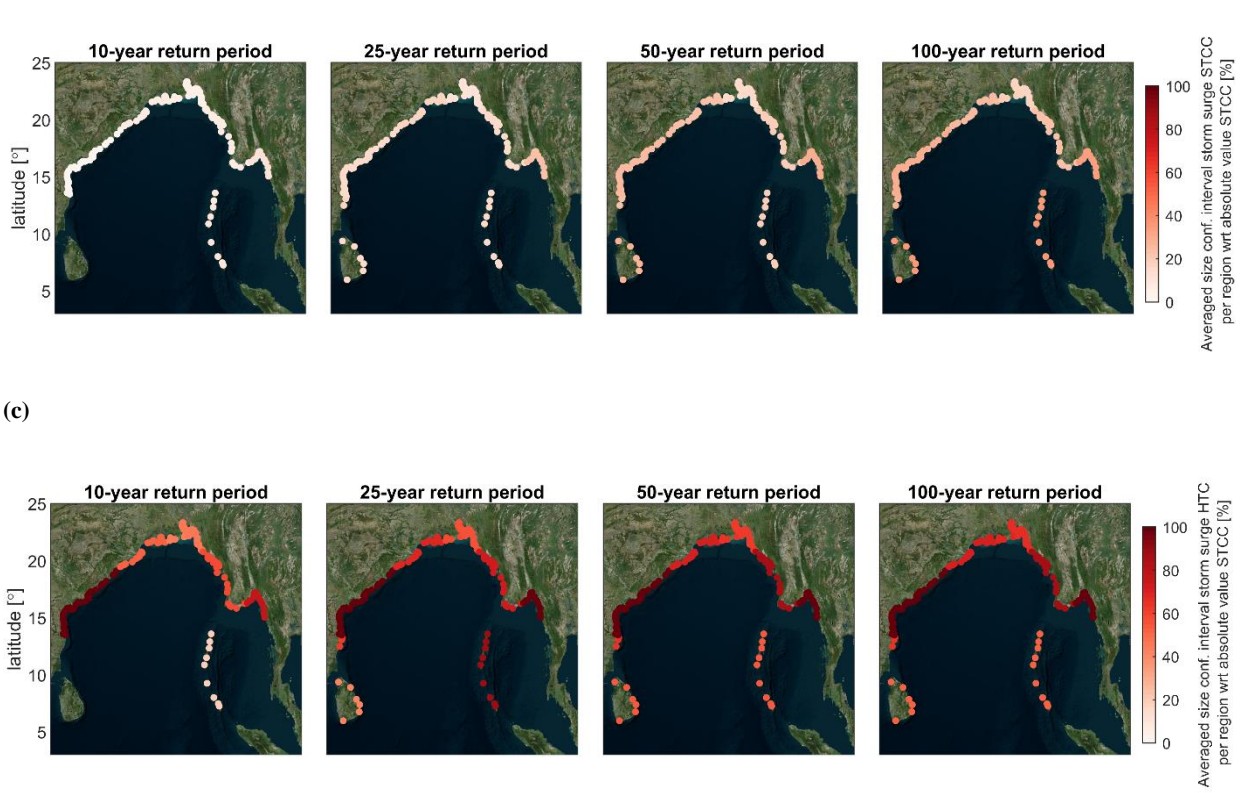

**(c)**

**(d)**

Figure 12 Panel **(a):** regionally averaged storm surges along the Bay of Bengal estimated based on STCC for return periods of 10, 25, 50 and 100 years. Panel **(b):** regionally averaged relative difference (in %) of storm surges estimated based on HTC compared to STCC of panel a. Panel **(c):** regionally averaged size of confidence interval of storm surge based on STCC as percentage (%) of absolute value of STCC of panel a. Panel **(d):** regionally averaged size of confidence interval of storm surge based on HTC as percentage (%) of absolute value of STCC of panel a.

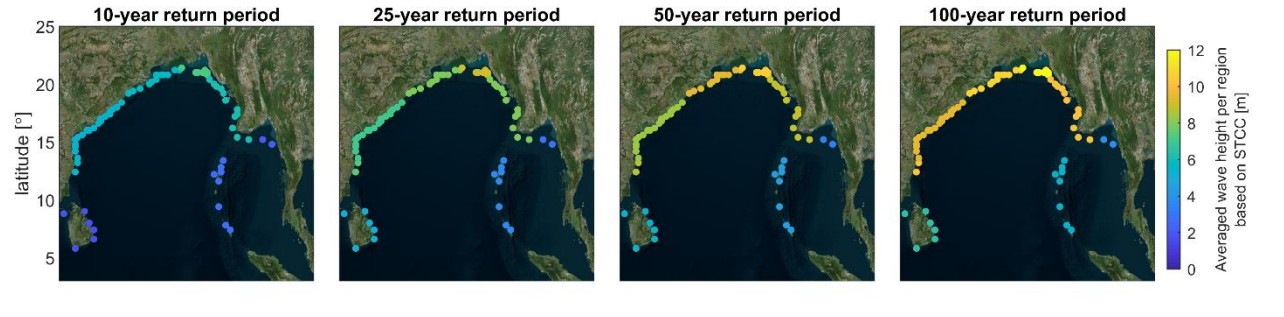

**(a)**
**(b)**

**(c)**

**(d)**

**Figure 13** Panel **(a):** regionally averaged wave heights along the Bay of Bengal estimated based on STCC for return periods of 10, 25, 50 and 100 years. Panel **(b):** regionally averaged relative difference (in %) of wave heights estimated based on HTC compared to STCC of panel a. Panel **(c):** regionally averaged size of confidence interval of wave heights based on STCC as percentage (%) of absolute value of STCC of panel a. Panel **(d):** regionally averaged size of confidence interval of wave heights based on HTC as percentage (%) of absolute value of STCC of panel a.




### 3.4 Effects of a changed future climate on storm surge and wave heights

The effect that a future climate with changes in cyclone wind speed (intensity) and frequency, possibly resulting from climate change, could have on resulting storm surge and wave heights was investigated using synthetic tracks (STCF, Section 2.3.1). Using the projected TC frequency and intensity changes of Knutson et al. (2015) for the NIO under RCP 4.5 for 2100, it was

quantified how changes in the first-order hazard (wind speed), could result into changes in the second-order hazards (storm surge and waves). In particular, it was estimated that, as a result of a TC frequency increase of 25.6% and an increase in maximum wind speed of 1.6%, this could lead to a relative increase in predicted storm surge of 11% and 6% respectively for return periods of 10 and 100 years and an increase of 9% and 6% for the wave height (see Figures 10 and 11 and Table 2). Therefore, the increase of storm surge and wave height could be larger than the increase in TC intensity only, but lower than

the increase in TC frequency, as a result of a combined effect resulting from an increase in TC intensity and frequency. If only the TC intensity increase would be relevant one would expect an increase of second-order hazards of 2.56%, since storm surge and wave heights are in the limit proportional to the wind speed squared. Besides, the CIs remain approximately the same as for the STCC case due to the already large number of samples for that scenario (see Figure B6 in Appendix B).

### 3.5 Summary of the results

To summarize the results, the values of Figures 12 and 13 are combined into Table 2, and also including the results estimated based on STCF. The values are presented for return periods of 10 and 100 years, both as averaged values over the entire BoB and per region. Included are the CI values (2.5th, 97.5th), indicating that the CI are significantly smaller based on synthetic tracks.

**Table 2** Storm surge and significant wave heights estimated based on STCC (black), HTC (**in bold**) and STCF (*in italic*) including confidence intervals (2.5[th], 97.5[th]), for return periods of 10 and 100 years, averaged over the entire BoB and per region. Regions are as in Table 1; Sri Lanka (whole country), India (TN = Tamil Nadu, AP = Andra Pradesh, OWB = Odisha & West Bengal and AN is Andaman and Nicobar Islands), Bangladesh (KB = Khulna & Barisal, CH = Chittagong) and Myanmar (RA = Rakhaing & Ayeyarwady, YM = Yangon & Mon). A '-' is displayed if no 10-year return period could be determined.





| Variable | Country / Region | Sri Lanka SL | India IN-TN | IN-AP | IN-OD | IN-AN | Bangladesh BA-KB | BA-CH | Myanmar MY-RA | MY-YM | Averaged over Bay of Bengal |
|---|---|---|---|---|---|---|---|---|---|---|---|
| Storm surge 10-year return period | STCC | - | - | 0.11 | 0.47 | 0.03 | 1.26 | 1.1 | 0.47 | 0.37 | 0.67 |
| | CI | (-,-) | (-,-) | (0.11,0.11) | (0.46,0.49) | (0.03,0.03) | (1.23,1.29) | (1.07,1.13) | (0.45,0.49) | (0.35,0.40) | (0.65,0.69) |
| | HTC | 0.05 | 0.11 | 0.17 | 0.38 | 0.03 | 0.96 | 1.03 | 0.35 | 0.44 | 0.46 |
| | CI | (0.04,0.07) | (0.08,0.14) | (0.11,0.25) | (0.27,0.51) | (0.03,0.04) | (0.71,1.26) | (0.79,1.29) | (0.24,0.51) | (0.33,0.62) | (0.34,0.62) |
| | STCF | - | 0.13 | 0.16 | 0.53 | 0.04 | 1.35 | 1.26 | 0.54 | 0.4 | 0.71 |
| | CI | (-,-) | (0.13,0.13) | (0.16,0.16) | (0.50,0.55) | (0.03,0.04) | (1.32,1.39) | (1.22,1.30) | (0.52,0.57) | (0.38,0.43) | (0.69,0.74) |
| Storm surge 100-year return period | STCC | 0.28 | 0.43 | 0.52 | 1.23 | 0.11 | 2.57 | 2.38 | 1 | 0.72 | 1.19 |
| | CI | (0.23,0.34) | (0.37,0.49) | (0.45,0.61) | (1.08,1.40) | (0.09,0.13) | (2.32,2.82) | (2.17,2.61) | (0.88,1.12) | (0.61,0.86) | (1.06,1.33) |
| | HTC | 0.12 | 0.28 | 0.52 | 0.89 | 0.06 | 2.09 | 2.25 | 0.81 | 0.82 | 1.04 |
| | CI | (0.05,0.20) | (0.16,0.43) | (0.22,0.91) | (0.49,1.35) | (0.04,0.10) | (1.30,3.04) | (1.51,3.07) | (0.44,1.29) | (0.50,1.33) | (0.62,1.56) |
| | STCF | 0.34 | 0.48 | 0.59 | 1.34 | 0.12 | 2.72 | 2.49 | 1.1 | 0.76 | 1.28 |
| | CI | (0.28,0.40) | (0.42,0.54) | (0.51,0.67) | (1.18,1.51) | (0.10,0.15) | (2.49,2.97) | (2.28,2.70) | (0.98,1.23) | (0.66,0.89) | (1.14,1.42) |
| Wave height 10-year return period | STCC | 1.71 | 5.58 | 5.39 | 5.75 | 2.79 | 6.98 | 7.14 | 6.11 | 2.17 | 5.45 |
| | CI | (1.69,1.72) | (5.47,5.69) | (5.31,5.47) | (5.60,5.92) | (2.74,2.83) | (6.86,7.10) | (7.01,7.27) | (6.00,6.22) | (2.13,2.21) | (5.35,5.56) |
| | HTC | 2.44 | 5.33 | 5.74 | 6.17 | 2.34 | 6.6 | 6.83 | 5.52 | 2.34 | 5.36 |
| | CI | (1.97,3.08) | (4.64,6.15) | (4.81,6.85) | (5.07,7.44) | (2.04,2.67) | (5.30,8.24) | (5.44,8.42) | (4.40,6.82) | (2.01,2.81) | (4.38,6.51) |
| | STCF | 3.27 | 6.14 | 5.96 | 6.59 | 3.05 | 7.52 | 7.67 | 6.72 | 2.38 | 5.97 |
| | CI | (3.20,3.33) | (5.97,6.31) | (5.84,6.09) | (6.41,6.77) | (2.97,3.13) | (7.36,7.70) | (7.50,7.85) | (6.56,6.89) | (2.31,2.46) | (5.83,6.12) |
| Wave height 100-year return period | STCC | 6.47 | 10.15 | 9.5 | 10.8 | 5.28 | 11.97 | 11.99 | 10.26 | 3.82 | 9.47 |
| | CI | (5.90,7.08) | (9.38,10.96) | (8.77,10.29) | (10.01,11.63) | (4.75,5.83) | (11.09,12.89) | (11.17,12.85) | (9.49,11.06) | (3.39,4.31) | (8.75,10.23) |
| | HTC | 4.55 | 9.43 | 10.77 | 11.97 | 3.87 | 12.07 | 12.73 | 10.49 | 4.08 | 10.04 |
| | CI | (2.83,6.89) | (6.84,12.56) | (7.29,14.93) | (8.38,16.11) | (2.79,5.06) | (8.58,16.48) | (8.97,17.03) | (7.18,14.39) | (2.85,5.84) | (6.97,13.68) |
| | STCF | 6.99 | 10.67 | 10.25 | 11.56 | 5.73 | 12.64 | 12.48 | 11.11 | 4.18 | 10.1 |
| | CI | (6.38,7.63) | (9.94,11.43) | (9.51,11.05) | (10.74,12.44) | (5.21,6.30) | (11.78,13.59) | (11.71,13.35) | (10.33,11.93) | (3.76,4.67) | (9.38,10.88) |



## 4 Discussion

For clarity, discussion points have been grouped under three main topics, namely: the generation of synthetic cyclone tracks, numerical modelling of storm surge and waves, and effects of a changed future climate on tropical cyclones.

### 4.1 Generation of synthetic tropical cyclone tracks

As shown by the reduction in the CI in the POT/GPD fit (see e.g. Figure 10), the uncertainty in modeling TC induced second-order hazards (i.e. storm surge and wave heights) is greatly reduced by using synthetic tracks. However, this is a reduction in
the uncertainty of the fitting parameters and thus estimates of return values and periods. Uncertainties regarding the wind parametrization and the correct representation of the climate in the underlying dataset of the TC persist. The former source of modelling uncertainties can be quantified by comparison with locally observed data. The latter, however, is a known unknown. TCWiSE is constrained to reproduce the statistics of the historical record. This means that the tool will not be able to (fully) reproduce physically-credible and statistically-unlikely tracks that are not recorded in observations. Additionally, in regions
of rare TC occurrence, the lack of multiple tracks in historical records as basis of the climate representation creates an unknown of how accurate the generated STC tracks represent the 'real climate' here. It will resemble the observed historical data with more realizations, where the historical data could in itself be biased within the limited time span of the observations compared to the 'real climate'. This cannot be verified but means that, in these regions, the results should be handled with care. However, for determining design criteria it is very common to use datasets of this (limited) time span.


Also, the STC have been generated using TCWiSE only once and used for both current and future climate conditions. When rerunning TCWiSE multiple times the generated tracks will be different (though with the same statistics). The effect of this difference in sampling has not been investigated. Additionally, using the HTC data, only one method to generate STC tracks was used while many more methods exist besides TCWiSE, e.g. Vickery et al. (2000), Hardy et al. (2003), James and Mason
(2005), Emanuel et al. (2006), Haigh et al. (2014), Lee et al. (2018) and Bloemendaal et al. (2020). Using these methods could potentially lead to different results, though it is expected that the main patterns and general conclusions will remain the same.

### 4.2 Numerical modeling of waves and storm surge

The currently used numerical models still have a relatively coarse resolution near the shore (> 2 km) and therefore are not capable of representing nearshore bathymetric features and their effects on nearshore wave conditions (e.g. wave shoaling,
refraction, breaking, etc.). The use of a higher resolution model would not be computationally feasible when run in combination with thousands of tracks and for a large domain as the BoB. For this reason, the estimated wave conditions were extracted at a water depth larger than 30 meters, under the assumption that waves would not be affected by bathymetric features at this depth.



As the scope of the paper lies on the comparison between hazards estimated by using historical and synthetic tracks and how the use of synthetic tracks can reduce the confidence intervals around the estimation of these hazards, the difference between simulated storm surge and wave heights versus observed values was not quantified.

Only wind and inverse barometric effect induced storm surge is considered for modeling extreme water levels, since tides are
not included. This is justified by the purpose of the paper (i.e. comparison between hazards induced by historical and synthetic tracks) and makes the comparison easier. Nevertheless, tidal effects would be important for the simulation of the total extreme water levels (see e.g. Chiu and Small, 2016). Additionally, river runoff due to extreme precipitation events could also increase local extreme water levels.

### 4.3 Effect changed future climate on tropical cyclones

The current approach of incorporating climate change induced effects via only including the change in TC intensity and frequency is still heuristic and limited in physical representation. Potential effects like changes in sea surface temperature that results in changes of the used statistics of locations of TC generation and termination, forward speed and heading are not incorporated yet. Moreover, a multi-model ensemble of different climate models feeding into the model train could result in more robust findings. However, the used cyclone frequency increase is similar to that of e.g. Sugi et al. (2017) that for the
NOI describes a 21% frequency increase for category 3-5 TC and also other literature describing similar trends (e.g. Walsh et al., 2016). At the same time, for low wind speed categories Sugi et al. (2017) found a reduction in cyclone frequency. Thereby, for specific regions/countries within the BoB the exact effects of climate change on TC are still argument of debate in literature and, so far, historical data does not seem yet to suggest any clear local trend. These predictions will also keep on being extended and improved, as well as extended scenarios with stronger climate change effects based on RCP 8.5 (e.g. Bhatla et
al. 2020; Knutson et al. 2019; Walsh et al., 2019).

### 5 Conclusions

In this study, estimates of extreme storm surge and significant wave heights induced by tropical cyclones were derived along the Bay of Bengal, both based on historical (deterministic method) and synthetic tracks (probabilistic method). Synthetic tropical cyclones tracks were generated by means of the TCWiSE tool for a period of 1,000 years using the statistics of
historical tropical cyclones as a basis but including a much larger number of realizations (i.e. 81 historical tracks versus 1745 synthetic tracks). It is shown that the statistics of the first-order hazard of wind speed are well reproduced by the synthetic tracks. Consequently, created wind fields were used to simulate the second-order hazards, namely storm surge and wave heights, based on the coupled process-based models Delft3D FM and SWAN. The study shows that, for the Bay of Bengal, about 400 years of synthetic results are required to reach convergence in results for wind speed, wave heights and storm surges,





with slight differences between the different variables. Since this is within 1,000 years, the synthetic tracks produce reliable estimates to compare the results based on historical tracks to.

An extreme value analysis performed over the computed storm surge and wave heights showed that, for the Bay of Bengal, the 95% confidence intervals using the synthetic tracks are 70-80% smaller than the confidence intervals estimated based on

historical tracks. The use of the deterministic method leads to an underestimation ranging between -31% and -13% for the estimated storm surge with return periods of 10 and 100 years and an under/overestimation between -2% and +6% for the wave heights for the same intervals. The use of synthetic tracks allows to better sample the full parameter space describing the tropical cyclones and to more accurately capture modelled extreme values, representing events with both more disadvantageous as advantageous trajectories which could statistically be plausible, but may not have happened yet historically. Hereby,

regional differences occur where regions in the south of the Bay of Bengal (e.g. Sri Lanka), that generally have a lower probability/numbers of historical cyclone events, show the largest underestimation of extreme waves computed based on the deterministic method compared to the probabilistic method. For extreme storm surge there is more variability from region to region, indicating that also in regions with a higher tropical cyclone probability using a deterministic method, could lead to an under- or overestimation of predicted values. The probabilistic method of deriving tropical cyclone induced coastal hazards

using synthetic tracks of TCWiSE can thereby improve the derivation of local design values anywhere in the world where tropical cyclone hazards exist.

Simulations were carried out both for current climate conditions as well as assuming changes in frequency and intensity of tropical cyclones, representing a future climate including possible effects of climate change. Literature values were used to

describe possible changes in tropical cyclone frequency and intensity by the year 2100 and there was modelled what this could imply in terms of changes in storm surge and wave heights. By assuming a possible increase in tropical cyclone frequency of +25.6% and tropical cyclone intensity of +1.6%, results show that this could result into an increase in the second-order hazard storm surge ranging between +11% and +6%, respectively for return periods of 10 and 100 years, and +9% and +6% for the wave heights. Thus, the combination of an increase in tropical cyclone frequency and intensity could result into a much larger

increase in second-order hazards (storm surges and wave heights) than the actual increase in tropical cyclone intensity only (first-order hazard wind speed), though lower than the increase in tropical cyclone frequency. However, the exact quantification of the effects of climate change on future tropical cyclones is still subject to debate and these differences are still smaller than the governing confidence intervals or differences in results when using a deterministic approach.

Since for local studies the followed approach of modeling thousands of synthetic tropical cyclones is (probably) not feasible considering that the (relatively coarse) numerical models of this study had a running time of about an hour per tropical cyclone, future work should focus on defining and reducing the needed number of synthetic tropical cyclone tracks.



## Appendix A – Used software

### TCWiSE

Code revision 66 of TCWiSE has been used, which is the same as described in Nederhoff et al. (2021). The open source tool
is available at the following link: https://www.deltares.nl/en/software/tcwise/.
Used settings:

- basinid = 'NI'
- dx = 0.1
- source = 'usa'
- window_KDE = 300
- window_dx = 15
- nyears = 1000
- exclude_land_map_KDE = 1
- methodlandv_KDE = 2
- deleteclosezeros_KDE = [1 1 0]
- merge_frac = 0.5
- dt = 3
- coupled_allowed = [0 0]
- decoupled_allowed = [1 1]
- latitude_allowed = [0 0 0]
- additional_landeffect = 1
- coefficientdecay = 0.0155
- termination_method = 1
- stochastic_radii = [0 0]
- wind_conversion_factor = 0.93
- cutoff_windspeed = 0
- cutoff_sst = 0

### 590 Delft3D Flexible Mesh Suite

The version of Delft3D FM used in this study is 1.2.8.62394. The Delft3D FM model is based on the grid of the GTSM (Muis
et al., 2016), with the coarsest resolution of 25 km on the ocean and the finest resolution along the coast of about 3 km.

The applied wind drag coefficient is not linearly increasing with wind speed. As described by Vatvani et al. (2012) the drag
coefficient first increases linearly up to a wind speed of 25 m/s (with $C_{d,max} = 0.005$), then it decreases linearly again up to a
wind speed of 50 m/s (with $C_d = 0.0025$). For even higher wind speeds the drag coefficient remains constant at this last value.

### SWAN

The version of SWAN used in this study is 40.91AB. For the SWAN model a rectilinear grid with a resolution of 0.2 degrees
was used (~2 km). The model was run in non-stationary mode with a time-step of 10 minutes. At the southern boundary at the
open ocean no information was applied and therefore all waves were generated internally as a result of the forced cyclones.





No background wind is included either for non-tropical cyclone conditions. A drag coefficient limiter was used with maximum value Cd,cap = 0.002 [-] as can be interpreted from (Zijlema et al., 2012) to limit the wave growth for very high wind speeds. For the whitecapping the formula of van der Westhuysen et al. (2007) was used. The bottom friction was set to a constant

bottom friction coefficient χ=0.038 m2s-3 as advised in (Zijlema et al., 2012).

Water levels and flow velocities were coupled with the wave model and updated every 30 min. The directional grid covers the full circle (360°), allowing for waves to travel to and from all directions. The number of directional bins was 36, which results in a directional resolution of 10°. The spectral grid covers a frequency range from 0.033 Hz to 0.5 Hz, allowing for wave periods from 2 to 33.3 s. The number of frequency bins is 30.

**Appendix B – Additional figures results**

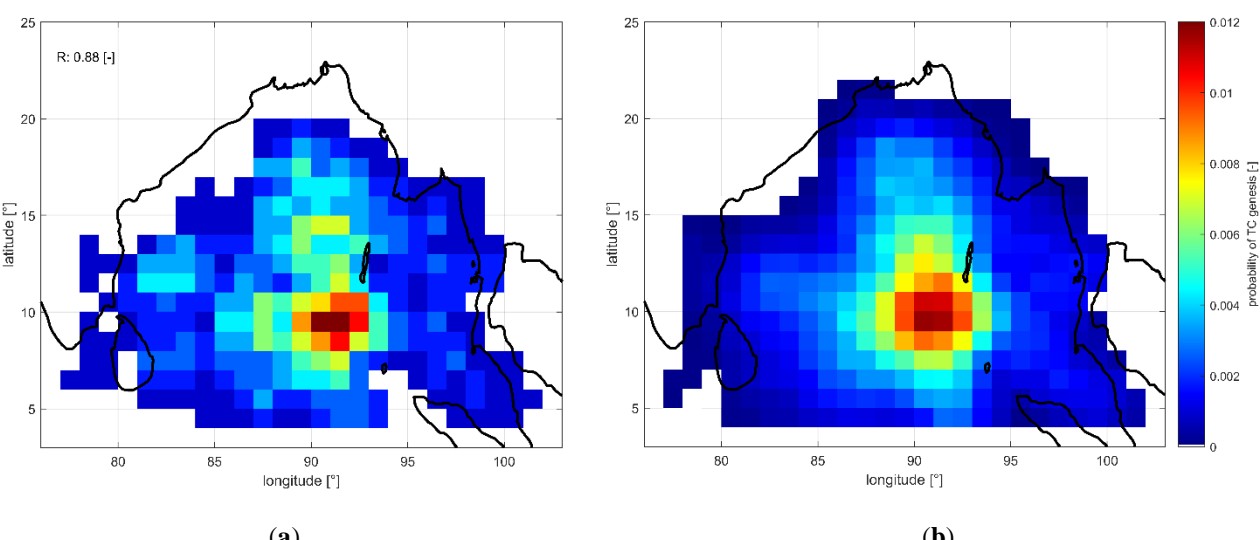

(a)                                                                    (b)



(c)

(d)

(e)

(f)

**Figure B1** Probability of TC genesis: (a) historical; (b) synthetic. Probability of TC termination: (c) historical; (d) synthetic. Yearly probability of a passing TC: (e) historical; (f) synthetic (STCC). Indicated error statistic is the correlation parameter R based on Kirchhofer (1974).



**Figure B**2 Comparison between CDFs of forward speed for HTC (black line) with 75% confidence intervals (dashed line) and STCC (blue line) at nine locations along the Bay of Bengal. The functions are estimated based on TCs within a 200 km radius from each location. The number of samples within the 200 km radius is indicated (#HTC and #STCC), alongside several statistical parameters comparing the HTC and STCC distributions (i.e. absolute difference in maxima (Δmax), normalized Mean Absolute Error (nMAE), the relative bias of the median value (bias), and the Root Mean Square Error (RMSE) of the whole CDF function. The nine locations are shown in Figure 1.

**Figure B3** Comparison between CDFs of heading direction for HTC (black line) with 75% confidence intervals (dashed line) and STCC (blue line) at nine locations along the Bay of Bengal. The functions are estimated based on TCs within a 200 km radius from each location. The number of samples within the 200 km radius is indicated (#HTC and #STCC), alongside several statistical parameters comparing the HTC and STCC distributions (i.e. absolute difference in maxima (Δmax), normalized Mean Absolute Error (nMAE), the relative bias of the median value (bias), and the Root Mean Square Error (RMSE) of the whole CDF function. The nine locations are shown in Figure 1.

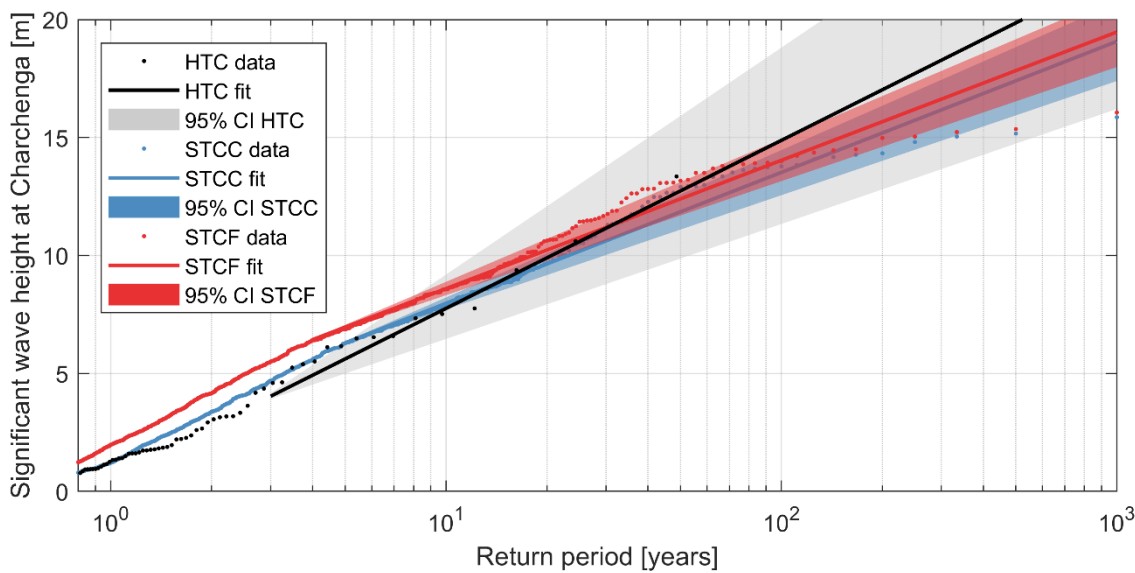

**Figure B4** Example of an Extreme Values Analysis for significant wave height at Charchenga, Bangladesh, for HTC (black), STCC (blue)
and STCF (red). The horizontal axis represents the return period in logarithmic scale, while the vertical axis represents the significant
wave height in meters. Shown are the data points with respective return periods (dots), the EVA fit (solid line) and the 95% confidence
intervals (colored fills).

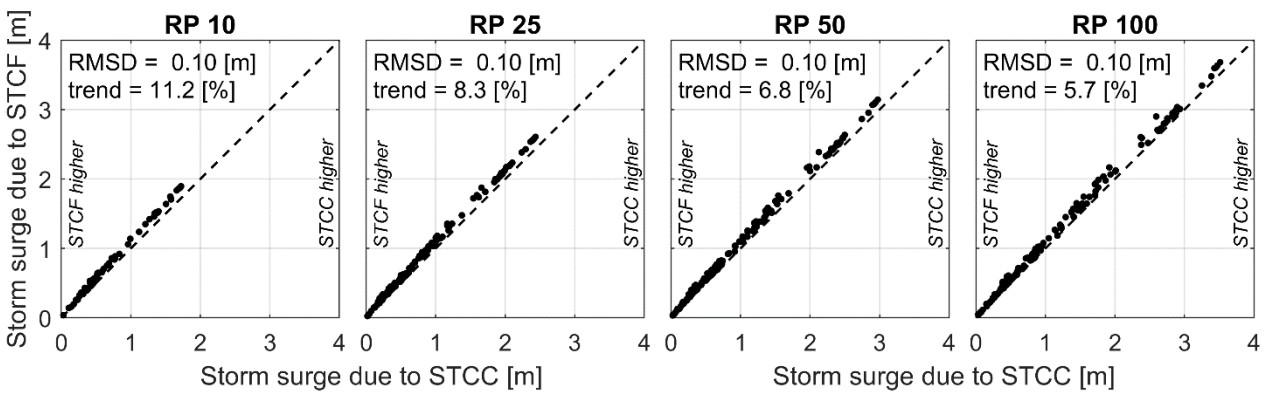

(**a**)




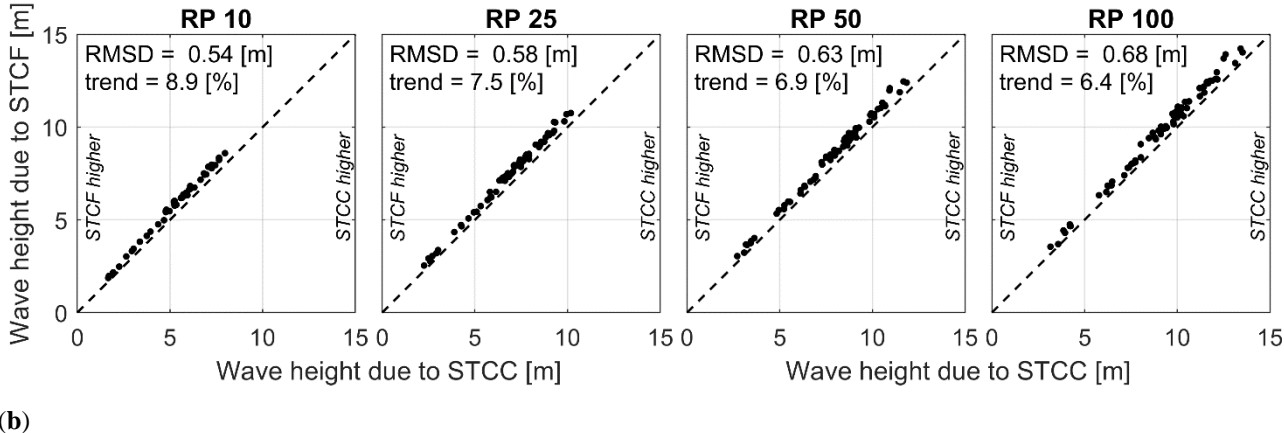

**(b)**

**Figure B5** Scatterplots of computed storm surge (panel a) and wave heights (panel b), both resulting from STCC (x-axis) and STCF (y-axis), for return periods of 10, 25, 50 and 100 years, for all locations along the Bay of Bengal. Root-mean-square differences (RMSD) and trend (%) of STCF compared to STCC are also shown.

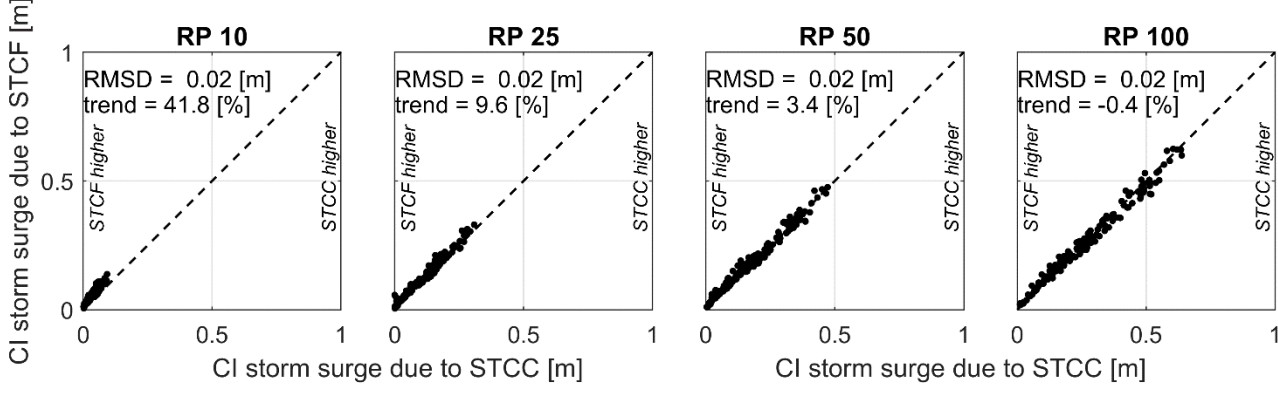

**(a)**





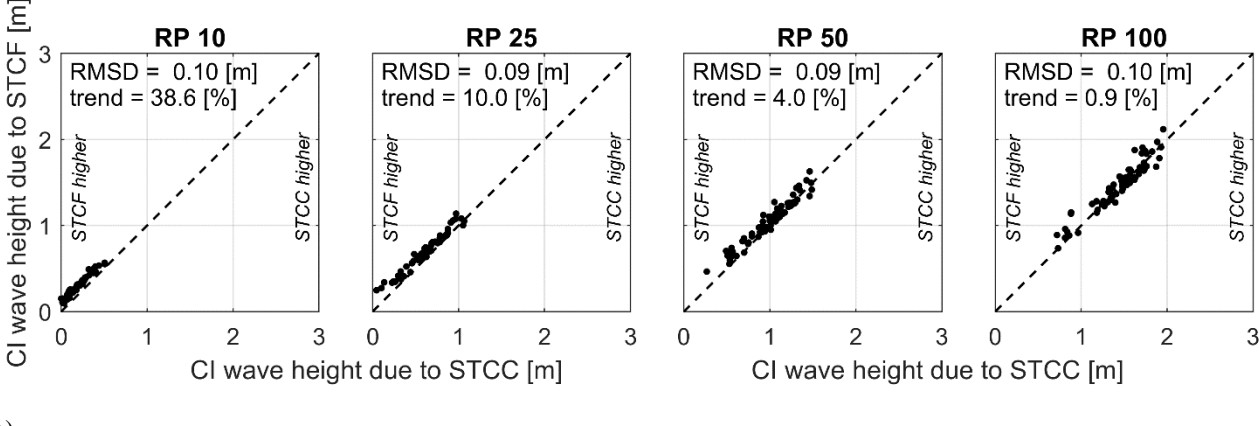

(**b**)

**Figure B6** Scatterplots of 95% confidence intervals (97.5th minus 2.5th percentile) for storm surge (panel a) and wave heights (panel b), both resulting from STCC (x-axis) and STCF (y-axis), for return periods of 10, 25, 50 and 100 years, for all locations along the Bay of Bengal. Root-mean-square differences (RMSD) and trend (%) of STCF compared to STCC are also shown.

**Code availability**

The ORCA toolbox to derive POT GPD is not open source but available through: https://www.deltares.nl/en/software/orca/

The TCWiSE tool is available through: https://www.deltares.nl/en/software/tcwise/

The Delft3D Flexible Mesh Suite is available through: https://www.deltares.nl/en/software/delft3d-flexible-mesh-suite/

**Data availability**

The IBTrACS dataset for HTC tracks is publicly available from: https://www.ncdc.noaa.gov/ibtracs/

**Author contribution**

All authors have read and agree to the published version of the manuscript. Conceptualization, methodology and writing—review and editing by all authors; software and resources by TL and KN; formal analysis by TL and SC; investigation, data curation and visualization by TL; writing—original draft preparation by TL and AG; funding acquisition by AG.

**Competing interests**

The authors declare that they have no conflict of interest.



**Acknowledgements**

We acknowledge the Deltares research programs 'Planning for Disaster Risk Reduction and Resilience', 'Seas and Coastal Zones' and 'Natural Hazards' which have provided funding to carry out the study and write the paper. Furthermore, we would like to thank Björn Röbke for helping in improving Figure 1.

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
