# Peer review of "Generating reliable estimates of tropical cyclone induced coastal hazards along the Bay of Bengal for current and future climates using synthetic tracks"

_Natural Hazards and Earth System Sciences, 2021_

## Author Comment (AC1)

**Rebuttal letter manuscript "Generating reliable estimates of tropical cyclone induced coastal hazards along the Bay of Bengal for current and future climates using synthetic tracks"**

Dear editor, dear reviewers,

On the June 21, 2021, we have submitted the following manuscript to the Journal of Natural Hazards and Earth System Sciences titled: "Generating reliable estimates of tropical cyclone induced coastal hazards along the Bay of Bengal for current and future climates using synthetic tracks" (MS No.: nhess-2021-181). On the October 21, 2021, we were informed that the open discussion was completed. In total, we received comments by two anonymous reviewers which provided positive feedback on the work done and valid suggestions. With this message we would like to acknowledge their time and efforts which we believe have led to an improvement to the quality and clarity of our manuscript. Below you will find a reply to all the specific questions and suggestions, which have also been addressed in the original manuscript.

Kind regards,

Tim Leijnse and co-authors
* * *
**Anonymous Referee #1**

This article provides an assessment for storm surge and wave hazards along the coast in the North Indian Ocean based on synthetic events. The method is based on synthetic tropical cyclones to provide more robust statistics compared to using only the short observational record on tropical cyclones in the area. I consider that the study is well structured, well presented and the science is sound, as such I recommend accepting the article after some minor revisions and one major are done.

*We thank the reviewer for the fast and thorough review, recommending our work for publication and providing useful feedback. Our response to the comments and how we have addressed them in the manuscript is given below in blue.*

Major comment:

The approximation for the future climate is based on Knutson et al. (2015), but in their work, they show a 200% increase in categories 4 and 5, which is not included in this work. The inclusion of the 200% increase for the most extreme events will likely change the results for the future climate, and I think this is a very important issue to discuss. Also, there is a new paper by Knutson et al. (2020), showing more details about the projections of TC into the future climate. While I do not suggest doing another assessment with the new projections, I do suggest a discussion of the selected projections in this work, with those reported in the latest work. I also suggest highlighting that the results presented for the future climate correspond to an RCP 4.5 scenario, although partially representing such scenario (as the 200% increase in cats 4-5 is not included), and also commenting that the results should be considered with caution as I believe this is more a proof of concept than an actual assessment for the motives explained above.

*Thank you for pointing this out. We consider this work indeed more as a proof of concept, a first step (see also L546-547) to understand how changes in TC might play out for coastal hazards. In the revised manuscript we are now emphasizing this further in the Abstract and Conclusions (L30-31; L586-58588), as well as part of the Discussion in Section 4.3. We refer now to the values of Knutson et al 2015 regarding the 200% increase for categories 4-5, noting however that this increase is marked statistically insignificant in that study (L75-77 & L539-L542). Additionally, we mention the changes interpreted for Knutson et al. 2020 (L544-546), although from their figures 1-3 it is hard to read exact numbers. We also emphasize that the results of our study based on one projection only (Knutson et al. 2015) should be interpreted with care as different projections may provide different values (L547-L548).*

*We have indeed not considered the work of Knutson et al. (2020), as our study was performed in 2019. However, in the revised manuscript, we have included a reference to it in the discussion (L538, 547-548).*

*We agree that it would be very interesting in follow-up research to be able to increase the frequency considering certain cyclone categories separately and improve the modelling of available future climate projections in general.*

Minor comments:

Lines 51-53: Please revise the sentence as it is not clear (...value analysis on than otherwise...)

*Thank you for noticing. We have adjusted L53-L54 and hope that it is clear now.*

Lines 56-58: I would suggest including the reference of Emanuel et al. (2008) and Lee et al. (2018) as they are those synthetic events are different in the sense that they are not prescribed with historical events climatology.

*Yes, these are good references. Lee et al. (2018) is mentioned in L57 & L502 and Emanuel et al. (2008) is mentioned in L57 & L503.*

Lines 59: The works by Meza et al. (2015) and Appendini et al. (2017) do consider the use of synthetic events to derive waves.

*Thank you for bringing this to our attention, we were not aware of these references. We have now added them and altered the statement, see L61-62;*

Line 72: Knutson et al. (2015) find an increase of 200% for category 4-5 storms, I think it is worth noticing such high value, which I do not think is reproduced by the synthetic events presented in this study. I think this is something to discuss later in the text.

*Good point. We have added this to L74-75 and, as mentioned above, is now also considered in the discussion, see Section 4.3 (L539-L542).*

Line 76: Please note that even without an increase in TC intensity, sea-level rise will lead to an increase of flood risk by TC.

*Yes, thank you for noting.  In the revised text we now state that an increase in TC intensity will amplify the flood risk further, in addition to the SLR effects (L79-80).*

Line 85: "This data can be used as boundary conditions", please clarify which is "this data"

*Text has been changed to "Derived estimates" (L88)*

Line 112: Please provide a brief explanation of the DIVA data for readers not familiar with it.

*Thank you for the suggestion. It is now being explained in L116-119.*

Lines 115-116: It is explained that the DIVA segments are translated to deepwater for waves, but what about storm surge? Storm surge is highly dependent on the bathymetry and slope so that it is relevant to provide the information related to the depth for the storm surge extraction, as well as a discussion of the accuracy.

*Indeed, as you stated, local storm surge levels might be strongly affected by bathymetry and slope. However, in this study we use the GTSM grid of Muis et al (2016), which has a nearshore resolution of 3 km (a resolution shown to lead to reasonably accurate results). The output locations were taken close to the shoreline but with no specification of a depth limit and at these locations the local bathymetry and slopes might not be fully resolved due to the relatively coarse model resolution of 3 km. This limitation of our study is now clearly stated in the discussion section, L507-509 and L512-514.*

Lines 171-176:The text is a bit confusing. As I understand, first the PDFs for the different variables is calculated based on historical, then a new PDF is calculated for maximum winds and increased by 1.6% to generate the future climate events, and then a subset of those events is selected for the present climate, for which the wind speeds are reduced by 1.6%. Please make the text clearer. Also, the work by Knutson et al. (2015) specifies an increase of 200% for category 4-5 storms, while this is not included, it would be good to discuss the implications it may have to omit this important value.

*Sorry for the lack of clarity. After the PDFs have been calculated based on the historical data these are not altered anymore. Only once the full tracks have been determined, the maximum wind speed of the track is increased by 1.6% in the future climate simulations. We hope that this is now more clearly explained in L176-180. The implications of considering other future climate increases are mentioned now in L 539-540.*

 Lines 214-215: The sentence is incomplete, please correct.

*We cannot follow this, maybe we should be looking at another line? In any case, we have removed the bracket in L220, but besides that did no further change/complete the sentence.*

Lines 360-362: It is not very clear this sentence. Why HTC are not considered "true values"?

*This is a valid question, and our point is that if historical records would have been much longer, they could be considered as true values. But since they only cover 48 years, we consider the record too short for fully reliable statistics. This can for instance be seen in Figure 6, where the spread of median estimated values for 50-year return values of included (synthetic) data is still very large. We try to clarify this in the manuscript with the sentence added in L370-372.*

Lines 379-380: Please specify which patterns and then include the section parenthesis so that the reader does not need to go back in the text.

*Sorry for being unclear. We have tried to improve the text following your suggestions, see L3904-391.*

Section 3.3.3. Some of the subplots in Figure 10 and most of them in Figure 11 show that both the STCC and STCF are embedded into the CI for HTC. I understand it is because the CI is too large for CI, but such a large CI for return periods below 50 years draws my attention. Could you please expand on this and its implications?

*The reason why the confidence interval (CI) are so large for the HTC data, even for relatively frequent events (e.g. return periods below 50 years), is the limited amount of historical data. If we look back at Figure 7, we saw that Charchenga has a relatively large cyclone probability: in the 48 years of historical data there are only 14 data points above the POT threshold, and only 4 datapoints with a return period larger than 10 years. This leads to quite some uncertainty in the CI. If we then look at locations in Figure 10 that have (significantly) lower cyclone probability (e.g. Batticaloa or Port Blair), one can imagine that here there are even fewer data points for the HTC, and that therefore the CI are very large. For wave heights this is a little bit less sensitive in the sense that swell waves generated further away can still reach the location, while storm surge is a more local effect. See also the discussion in L395-399.*

Section 3.4. As I noted above, I think it is needed to highlight that Knutson et al. (2015) found an increase of 200% for categories 4-5, which is not considered in this study. Also, it needs to be noted that such results are based on an RCP 4.5, so the results in this work also represent that scenario.

*As noted above, added to section 4.3, L539-542. RCP 4.5 is mentioned in L72, 175 & L456.*

Section 4.1. I suggest that you discuss the last three sentences in relation to physics-based synthetic events, such as the ones from Emanuel (2008).

*We have added a sentence, see L503.*

Section 4.2. There is no discussion on the effect of bathymetry, which also has a large repercussion on storm surges. Please also discuss those limitations. I would suggest that at some point in the paper it is specified that the values obtained, at least for storm surge, need to be considered as indicative to provide a comparison between historical and synthetic (present and future climates), but the values per se need to be handled with care, as there is no validation or calibration of the wave and storm surge results.

*Yes, we fully agree. This paper should be seen as a proof of concept and do not represent values to be used directly in detailed design. We added this in the discussion (L518-520). Also, in the revised version of the manuscript, text has been added on the storm surge extraction depth (L512-514).*

Lines 514-516: This is not clear in the study as an overall increase frequency was applied, as well as increased intensity, based on Kuntson (2015). As such, the increase in 3-5 categories such as found in Sugi et al. (2017) (and 200% increase found by Knutson et al. 2015) is not really incorporated in this study.

*Indeed, these values have not been incorporated in the study, they were mentioned as indicative. In the revised manuscript we clearly state and motivate the used values from Knutson et al 2015 and discuss other literature in L589-548.*

Line 537-538: "The use of synthetic tracks allows to better sample the full parameter space describing the tropical cyclones and to more accurately capture modelled extreme values". The "better sample" part is related to lower CI, right? I think it should be put into that context. Maybe it is better to use the word robust instead of better.

*The "to better sample" refers to sampling many more different combinations of landfall location/heading/forward speed/wind speed and thereby better describing the parameter space by mean of continuous PDFs. This results in lower confidence intervals. We replaced better with robust in L533.*

Lines 560-562: It could be added that as an alternative to reducing the number of synthetic events, the application of fast algorithms to determine storm surges and waves could be another possibility.

*Thank you for the suggestion, which we have incorporated this in L592-593.*

Table 2: While the information is relevant, it would be easier to see the graphically with bar plots. For instance a figure with subplots, where each is a return period. On each subplot, all the locations are shown with bars for each dataset. The bars could even show the CI as error bars. I think this would allow the reader to see more clearly the data.

*We prefer to keep it as it is. Most of these data are already shown in the separate subplots of figures 12 and 13. Therefore, in our opinion there is more added value in specifying the numbers as done in Table 2, than adding extra plots.*

Figure 4: Considering that this figure corresponds to the methodology for generating the synthetic events, I would consider that a time series of wind speed would be more adequate. Also, I suggest showing a plot with the PDFs for each database. While the CDF is presented for particular locations, comparing historical vs synthetic PDFs or CDFs for the entire basin would provide a measure of the method robustness, which is not necessarily shown for a particular location (that is discussed later as an area not experiencing yet a specific track, such randomness is diffused when considering the entire basin)

*With relation to the first point, our goal is solely to visualize the length of the time-series, and not the values themselves. The current figure is therefore deemed sufficient and has not been adjusted.*

*With relation to the second point, in Appendix B spatial plots of genesis/termination/yearly probability are shown including error statistics. Combined with the showed CDFs of figures 5,B2,B3 that span 9 locations across the entire coastline, this provides in our opinion enough indication of the methods robustness. For completeness, we have included figures of heading, forward speed, maximum sustained winds to this rebuttal below. Note that the vmax synthetic plot here includes the intensity increase of future climate. As the figures show, when focusing on the ocean part of the Bay of Bengal (tracks not reliable on land), there is a general agreement between the patterns.*

*Heading*

[Figure]

*Forward speed*

[Figure]

*Vmax*

[Figure]

Figures 10 and 11: Why do the values start on different return periods for each location? Is it a result of the threshold used in POT? For instance, Port Blair in Figure 10 starts with a 40 years return period for STCC, could that be a result of too few data points from the POT to perform the EVA? It makes me wonder if the POT criteria is correct for some stations.

*This is indeed because of the POT threshold having different values for HTC/STCC/STCF. Section 2.3.4 describes the method in more detail, but these are based on percentile values between 98-99.9 for HTC and 99-99.9 for STCC/STCF. Thereby a stable region for the shape parameter of the Pareto distribution is found, ensuring stable fits. Since there are more data available for the synthetic tracks, these percentile values lead in general to higher thresholds, e.g. as in Figure 7, and higher accuracy than the estimates from the HTC data, which for some of these locations the sample sizes are rather low. See figures below that show this for the EVA of the water levels at Port Blair. The individual fits seem appropriate even though the sample sizes are low.*

*HTC*

[Figure]

STCC

*STCF*

[Figure]

References:

Appendini, C. M., Pedrozo-Acuña, A., Meza-Padilla, R., Torres-Freyermuth, A., Cerezo-Mota, R., López-González, J., & Ruiz-Salcines, P. (2017). On the Role of Climate Change on Wind Waves Generated by Tropical Cyclones in the Gulf of Mexico. Coastal Engineering Journal, 59(2), 1740001-1-1740001–32. https://doi.org/10.1142/S0578563417400010

Emanuel, K., Sundararajan, R., & Williams, J. (2008). Hurricanes and Global Warming: Results from Downscaling IPCC AR4 Simulations. Bulletin of the American Meteorological Society, 89(3), 347–367. https://doi.org/10.1175/BAMS-89-3-347

Knutson, T., Camargo, S. J., Chan, J. C. L., Emanuel, K., Ho, C.-H., Kossin, J., et al. (2020). Tropical Cyclones and Climate Change Assessment: Part II: Projected Response to Anthropogenic Warming. Bulletin of the American Meteorological Society, 101(3), E303–E322. https://doi.org/10.1175/BAMS-D-18-0194.1

Lee, C. Y. Y., Tippett, M. K. K., Sobel, A. H. H., & Camargo, S. J. J. (2018). An environmentally forced tropical cyclone hazard model. Journal of Advances in Modeling Earth Systems, 10(1), 223–241. https://doi.org/10.1002/2017MS001186

Meza-Padilla, R., Appendini, C. M., & Pedrozo-Acuña, A. (2015). Hurricane-induced waves and storm surge modeling for the Mexican coast. Ocean Dynamics, 65(8), 1199–1211. https://doi.org/10.1007/s10236-015-0861-7

*Thank you for these. As indicated in the replies above, we have added these references where appropriate.*

---

## Author Comment (AC2)

**Rebuttal letter manuscript "Generating reliable estimates of tropical cyclone induced coastal hazards along the Bay of Bengal for current and future climates using synthetic tracks"**

Dear editor, dear reviewers,

On the June 21, 2021, we have submitted the following manuscript to the Journal of Natural Hazards and Earth System Sciences titled: "Generating reliable estimates of tropical cyclone induced coastal hazards along the Bay of Bengal for current and future climates using synthetic tracks" (MS No.: nhess-2021-181). On the October 21, 2021, we were informed that the open discussion was completed. In total, we received comments by two anonymous reviewers which provided positive feedback on the work done and valid suggestions. With this message we would like to acknowledge their time and efforts which we believe have led to an improvement to the quality and clarity of our manuscript. Below you will find a reply to all the specific questions and suggestions, which have also been addressed in the original manuscript.

Kind regards,

Tim Leijnse and co-authors
* * *
**Anonymous Referee #2**

***We thank the reviewer for the review, in particular for the useful feedback and interesting references. Our response to the comments are given in blue below.***

Literature reviews are not enough:

For STC, Nakajo et al. (2014) is preferable.

Nakajo, S., N. Mori, T. Yasuda and H. Mase (2014) Global stochastic tropical cyclone model based on principal component analysis with cluster analysis, Journal of Applied Meteorology and Climatology, American Meteorological Society, Vol.53, pp.1547-1577.

*Thank you for bringing these references to our attention. Nakajo et al. has been added to the list of STC methods mentioned in the manuscript, L57 & L502.*

For the second-order hazard due to TC, the paper should include recent studies using datasets of a STC model and climate change experiments, for example, the following references:

Yasuda, T., S. Nakajo, S. Kim, H. Mase, N. Mori and K. Horsburgh (2014) Evaluation of Future Storm Surge Risk in East Asia based on State-of-the-art Climate Change Projection, Coastal Engineering, Volume 83, January 2014, Pages 65–71

Mori, N. and T. Takemi (2016) Impact assessment of coastal hazards due to future changes of tropical cyclones in the North Pacific Ocean, Weather and Climate Extremes (review paper), Vol.11, pp.53-69. doi: 10.1016/j.wace.2015.09.002

Mori, N., M. Kjerland, S. Nakajo, Y. Shibutani and T. Shimura (2016) Impact assessment of climate change on coastal hazards in Japan (review paper), Hydrological Research Letters, Vol.10(3), pp.101-105. doi: 10.3178/hrl.10.101

Yang, J.A, S.Y. Kim, N. Mori, H. Mase (2018) Assessment of long-term impact of storm surges around the Korean Peninsula based on a large ensemble of climate projections, Coastal Engineering, Elsevier, Vol.142, pp.1-8. doi.org/10.1016/j.coastaleng.2018.09.008

Mori, N., T. Shimura, K. Yoshida, R. Mizuta, Y. Okada, M. Fujita, T. Temur Khujanazarov, E. Nakakita (2019) Future changes in extreme storm surges based on mega-ensemble projection using 60-km resolution atmospheric global circulation model, Coastal Engineering Journal, Taylor & Francis, 61(3), pp.295-307.

Yang, J.A, S.Y. Kim, S.Y. Son, N. Mori, H. Mase (2020) Assessment of uncertainties in projecting future changes to extreme storm surge height depending of future SST and greenhouse gas emission scenarios, Climatic Change, pp.1-18. https://doi.org/10.1007/s10584-020-02782-7

Sooyoul Kim, Jihee Oh, K.D. Suh and H. Mase (2017) Estimation of climate change impacts on storm surge: Application to Korean Peninsula, Coastal Engineering Journal, 59, 170004, 10.1142/S0578563417400046.

*Thank you for these interesting references. We now refer to many in L62 and L503 (Mori et al 2016a, Mori et al 2016B, Mori et al 2019, Yang et al 2020).*

In 55, the authors mention "local design values" for wave heights. But you investigate them in the relatively deep water. What kind of "local design" is the purpose? It should be clear.

*Thank you for pointing this out. We do indeed determine the wave heights in relatively deep water. Ideally, we would have done that in shallower waters, what we would like to do in the future, but that was now not yet computationally feasible to do for the entire BoB. We have changed the text now to 'offshore extremes', L59.*

*Also, in the discussion Section 4.2 we now emphasize that our model results are indicative and should not be directly used in detailed design L518-520.*

When we say a "coupled model", we mention physical processes through coupling. How did you make the coupled model physically?

*The coupled Delft3D FM-SWAN model exchanges water levels and radiation stresses every hour, we have added this to the text in L196-198. Thank you for bringing this addition to our attention.*

When we consider waves in storm surges, we think of radiation stress in the momentum equations for storm surges. Also, we can consider wave runup/overtopping for coastal floods. Why did you consider waves / why did you use a coupled model in your study? Is any typical reason for it?

*Good question. We included waves in our study because besides the effects of HTC vs STC tracks on storm surge, we wanted to evaluate the effect on wave heights too. Thereby it was most straight-forward to do that all in 1 model suite (that of Delft3D) where both models run coupled, with as benefit that effects of the waves on storm surge (and vice-versa) as the reviewer mentions are included too. We have added text on the exchange of water levels and radiation stresses in L198.*

In 140, why did the authors choose POT for statistical analysis? How did you determine the threshold value for each station/location/region? Did you consider other methods, likely the annual maximum series? Why did you use a Generalized Pareto Distribution? Is it a representative in this region?

*Sorry for not having motivated it. We apply the POT approach because of the available sample sizes. The Generalized Pareto Distribution is fitted because it is the asymptotic distribution of the POT data. The POT thresholds are determined per station based on minimum percentiles and using the threshold stability criteria of Caires 2016 (see explanation in section 2.3.4). Annual maximum series were not used because at most locations in the BoB, the yearly probability of a cyclone is below 1.*

In 3.1, the authors investigate the wind speed. But I am surprised why the pressure/central pressure of TC is not studied. The driving force of the wave is absolutely the wind speed. BUT that of the storm surge is the wind speed and the pressure of TC. Therefore, the PRESSURE has to be verified for discussing the storm surge. It is the most significant lack point in this paper. Without discussing the central pressure of TC, the discussion of the future change of storm surges has no meaning.

*Thank you for pointing this out and our excuses for the confusion. As you rightly state the pressure is a driver and important when assessing the impact of TCs and is fully accounted for in our computations.*

*For the verification of the synthetic TC tracks in section 3.1 the pressure was not investigated specifically, since pressure is not one of the variables that is directly sampled in TCWiSE. Since wind speed and central pressure of a TC are directly correlated, TCWiSE samples values of wind speed as proxy for intensity, where after using Holland et al. 2008 a corresponding pressure is found. This is done when creating the 2D pressure fields, as explained in section 2.3.2. To show that central pressure (as derived value from wind speed through Holland 2008) is also estimated fairly well we have added Figure B4 in the appendix showing the CDFs for the 9 locations. Error statistics are added to the main text in L268-271.*

*Furthermore, sorry for not stating it clearly, but the central pressure is included in our storm surge modelling. In Delft3D both the wind speed and atmospheric pressure are included as 2D fields which drive the storm surge. We hope that the revised manuscript clarifies your question (L 201-202).*

*Since we derive the pressure from the wind speed and apply it in calculating the storm surge in Delft3D, the influence of the TC pressure and the possible changes in the pressure with climate*

*change (derived from the increase in wind speeds) is fully accounted for in our study when calculating the future climate results. Possible changes in the relationship between TC central pressure and wind speed as a result of climate change are not included (since we use the empirical relationship of Holland). To highlight this, we have included this as a discussion point in Section 4.3, see L 531-532.*

In 275, the validation process has to include the effect of the central pressure of TC on the storm surge.

*We agree that the pressure fields are an important forcing for storm surge computations and have added Figure B4 in the appendix and error statistics in the main text as mentioned above.*

In 390, I disagree with these words because the central pressure is omitted.

*We hope to have clarified, through our replies above and adjustments and additions to the manuscript, that central pressure is not omitted. In the modelling of storm surge in Delft3D, both the wind speed and the central pressure are used to determine the storm surge.*

---

## Author Response (AR2)

**Response to the editor:**

Dear Piero,

Thanks a lot for evaluating the revisions of the paper and for recommending our article for publication.

To explain on the minor clarification on the terminology, we call storm surge and waves second order hazards because they are generated by and a consequence of the wind, which we thereby call the first order hazard.

To make this clearer in the manuscript, we have added this both to the abstract as the introduction:

*L20 "Based on estimated wind fields, second-order hazards (i.e. storm surge and waves) are computed that are generated by the first-order hazard of wind."*

*L56 "Subsequently, these STC can then be modelled using hydrodynamic and wave models to generate better estimates of second-order hazards like storm surge and wave heights that are generated by the first-order hazard of wind."*

Additionally, we have added a link to Zenodo where a dataset of our results can be downloaded by readers, with proper reference there to the NHESS article DOI.

Thank you for considering our submission,

Tim